# Event-based Probabilistic Risk Assessment of Livestock Snow Disasters in the Qinghai-Tibetan Plateau

Tao Ye[1,2,3], Weihang Liu[1,4] Jidong Wu[1,2], Yijia Li[1,2*], Peijun Shi[1,2] and Qiang Zhang[1,2]

[1]State Key Laboratory of Earth Surface Processes and Resource Ecology, Key Laboratory of Environmental Change and Natural Disaster, Ministry of Education, Faculty of Geographical Science, Beijing Normal University, Beijing 100875, China;
[2]Academy of Disaster Reduction and Emergency Management, Faculty of Geographical Science, Beijing 100875, China
[3]Frederick S. Pardee Center for the Study of the Longer-Range Future, Boston University, Boston 02215, the U.S.
[4]School of Geographic Science, East China Normal University, Shanghai 200241, China

*Correspondence to*: Yijia Li(liyijia@mail.bnu.edu.cn)

**Abstract.** Understanding risk using quantitative risk assessment offers critical information for risk-informed reduction actions, investing in building resilience, and planning for adaptation. This study develops an event-based probabilistic risk assessment model for livestock snow disasters in the Qinghai-Tibetan Plateau (QTP) region and derives risk assessment results based on historical climate conditions (1980–2015) and present-day prevention capacity. In the model, a hazard module was developed to identify/simulate individual snow disaster events based on boosted regression trees. Combining a fitted quantitative vulnerability function and exposure derived from vegetation type and grassland carrying capacity, we estimated risk metrics based on livestock mortality and mortality rate. In our results, high risk regions include the Nyainqêntanglha Range, Tanggula Range, Bayankhar Mountains and the region between the Kailas Range and neighbouring Himalayas. In these regions, annual livestock mortality rates were estimated as > 2% and mortality was estimated as >2 sheep unit/km² at a return period of 20-year. Prefectures identified with extremely high risk include Guoluo in Qinghai Province and Naqu, and Shigatse in the Tibet Autonomous Region. In these prefectures, a snow disaster event with return period of 20-year or higher can easily claim total losses of more than 500,000 sheep units. Our event-based PRA results provide a quantitative reference for preparedness and insurance solutions in reducing mortality risk. The methodology developed here can be further adapted to future climate change risk analyses and provide important information for planning climate change adaption in the QTP region.

## 1 Introduction

Livestock snow disasters are serious winter extreme weather events that widely occur in central-to-east Asian temperate steppe and alpine steppes (Li et al., 2018; Tachiiri et al., 2008). In the pastoral areas of these regions, heavy snow fall provides thick and long-lasting snow cover, making forage unavailable or inaccessible (Fernández-Giménez et al., 2015). Together with extremely low temperature and strong wind, this cover severely inhibits natural grazing, claims considerable livestock mortality, and brings devastating impacts to the livelihoods of local herders, even threatening their survival (Wang et al., 2013a). In response to threats from livestock snow disasters, great efforts have been devoted to understanding their mechanism as a complicated interaction between precipitation, vegetation, livestock, and herding communities (Nandintsetseg et al., 2018;

Shang et al., 2012; Sternberg, 2017); the major drivers of (socioeconomic) vulnerability (Fernández-Giménez et al., 2012; Wang et al., 2014; Wei et al., 2017; Yeh et al., 2014); and key factors that could foster adaptive capacity and community resilience (Dong and Sherman, 2015; Fernández-Giménez et al., 2015). Attempts have been made to develop techniques, such as snow disaster monitoring, forecasting, and rapid assessment, to provide critical information for prevention and addressing emergencies (Wang et al., 2013b; Yin et al., 2017). Quantitative analyses have also been conducted to derive the relationship between livestock loss, snow hazard, and various environmental stressors (Li et al., 2018; Mukund Palat et al., 2015).

Disaster risk is a measure of uncertain consequences. The Sendai Framework outlines the importance of risk assessment as a critical means of understanding disaster risk and a prerequisite for other actions, e.g., risk-based investment for resilience and adaptation (UNISDR, 2015). Following the mainstreaming risk assessment framework of Risk = Hazard × Exposure × Vulnerability (Jongman et al., 2015), several ordinal risk assessment studies have evaluated livestock snow disasters in Inner Mongolia and the Qinghai-Tibetan Plateau (QTP) of China (Li et al., 2014; Liu et al., 2014a; Wu et al., 2007). Generally, they derive the measure of risk as an ordinal index by integrating indices representing different components of risk, e.g., the world risk report (Birkmann and Welle, 2016). This ordinal risk assessment approach offers only rankings but no quantitative information of underlying risk, i.e. the uncertainty of consequences. Consequently, it can be valuable for policy-making but provides little support for risk-informed decisions, e.g. insurance pricing or cost-benefit analysis.

Researchers have also derived probabilistic risk assessment (PRA) results for livestock snow disaster. In such a framework, risk measured as a probability distribution of socioeconomic losses (consequences) are generally derived with the probability distribution of hazard intensity and dose-response relationships between hazard intensity and socioeconomic losses (Carleton and Hsiang, 2016; Michel-Kerjan and Kousky, 2010; Shi and Kasperson, 2015). Bai et al. (2011) applied the PRA framework to a livestock snow disaster risk assessment in Qinghai Province of China. A function of livestock mortality rate in response to the snow season (November to April of the preceding year) daily average snow depth using historical disaster records. Historically annual average snow depth computed from satellite-retrieved data were used to derive return-period livestock mortality and mortality rates as the final risk metrics. Based on their method, quantitative livestock snow disaster risks were mapped nationwide in China (Shi, 2011). Tachiiri and Shinoda (2012) successfully extended the framework to future climate change analysis. They trained a tree-based model to link annual livestock loss rates, the October to April snow water equivalence, and normalized difference vegetation index. The projected snow water equivalence values from climate scenarios were then used to estimate the frequency of anomalous livestock loss rates >5% or >17% for 2010–2099. Ye et al. (2017) further extended the PRA framework to support insurance design and pricing using snow season cumulative snow-cover days. These earlier studies primarily developed their PRA models using annual variables. In this study, we develop an event-based PRA method for present and future livestock snow disaster risk assessments for the QTP region. The event-based PRA approach has several important additions compared to earlier studies using annual variables. 1) From the modeling perspective, the event-based framework retains the capability to accommodate multiple events in a year, which is one characteristic of snow disasters. This is important for snow disaster as earlier studies has demonstrated that livestock mortality rate exhibits a concave relationship with disaster duration (Li et al., 2018). The losses of one event lasting for 30 days and two-events lasting for 15 days each are clearly different. In addition, modelling events provides a mechanism for capturing the change in event frequency

and intensity in response to environmental change, such as climate change. 2) From a risk-informed action perspective, the annual evaluation, i.e., potential aggregate duration, is not useful for risk-transfer mechanisms because insurance needs to address the natural event basis, although it might be temporarily acceptable for annual preparedness planning. This is also the critical reason that catastrophe risk models are mostly built on an event-basis (Michel-Kerjan et al., 2013).

There are three major aims of this study: 1) Develop a hazard module that can identify/capture snow disaster event based on daily weather data. It is the basis for any event-based modelling attempts, and is particularly important for regions where historical records are absent and for future risk assessment where observations and records are not yet available and variabilities from future climate change will exist. 2) Set up an event-based PRA framework for livestock snow disaster risk assessment by integrating snow disaster event (hazards), livestock vulnerability, and exposure together to derive a probabilistic quantification

of risk. 3) Derive the risk metrics for livestock mortality risk in the QTP and offer risk-informed reduction implications. Worldwide, the QTP is a region that has most-suffered from livestock snow disasters due to its large snow cover area, long-lasting snow cover days, and nomadic grazing (Li et al., 2018). This region is also a hot spot in climate change (Diffenbaugh and Giorgi, 2012; Gu et al., 2014). Quantitative risk assessments for the present day will likely be a significant source of information for disaster risk reduction. In addition, our framework can be adapted for livestock mortality in snow disasters in

the context of future climate change analysis, and therefore support climate adaptation planning for local government and herding communities.

## 2. Materials and methods

### 2.1 Study area

The QTP contains the world's highest elevation pastoral area (Wang et al., 2016). It has extremely enriched grassland resources,

with a total alpine grassland cover of $1.57 \times 10^6$ km$^2$, supporting the livelihood of approximately 2 million pastoralists and 3 million agro-pastoralists (Miller, 2005). In 2014, the QTP housed a total of 38.03 million livestock, and animal husbandry production reached 23.85 billion RMB yuan[1]. Typical nomadic grazing has been used for centuries, and today it remains the most popular way of raising livestock (Wang et al., 2014). Local herders rely heavily on open-air free grazing and possess poor infrastructure, such as thermal sheds, and rarely prepare hay and fodder for potentially harsh winters. Provincial and local

governments have been investing to improve prevention capacity and snow disaster resilience for local communities, but further efforts are still needed to reach a satisfactory solution due to economic and conventional constraints (Shang et al., 2012; Ye et al. 2018).

The Tibetan Plateau is one of the three primary snowfall regions in China (Yin et al., 2017; Qin et al. 2015). On average, the snow cover can attain $0.61 \times 10^6$ km$^2$ in the winter season (Duo et al., 2014) and persist for over 240 days (Basang et al., 2017).

The large snow cover area, long-lasting snow cover days, and nomadic grazing make this area suffer from livestock snow disasters more than other regions. A total of 18 million livestock died in snow disasters during the 1974–2009 period in the

---

[1] 1 yuan = 0.146 USD as of Dec 27, 2018

eastern Tibetan Plateau (Wang et al., 2016). In the 1995–1996 snow season, consecutive snow disaster events killed 1.29 million livestock (Wen, 2008). The 1997–1998 snow season in Naqu, central Tibetan Plateau, led to the loss of 0.82 million livestock, and threatened the lives of 100,000 local residents (Wen, 2008b).

## 2.2 Methods

In this study, an event-based PRA framework was developed for livestock snow disaster (Fig. 1). We followed the PRA approach proposed by Carleton and Hsiang (2016), and applied the concept of event-based modelling in catastrophic risk models (Michel-Kerjan et al., 2013). The event-based modelling approach was framed using state-of-the-art three-element risk modelling, hazard, exposure, and vulnerability (Kinoshita et al., 2018; Muis et al., 2015) to model losses claimed by individual events. Then PRA was achieved through repetition of individual event modelling, in which a large number of events were

drawn from the full distribution of hazards, given the predicted losses/consequences from individual events, from which a full distribution of disaster loss can be obtained.

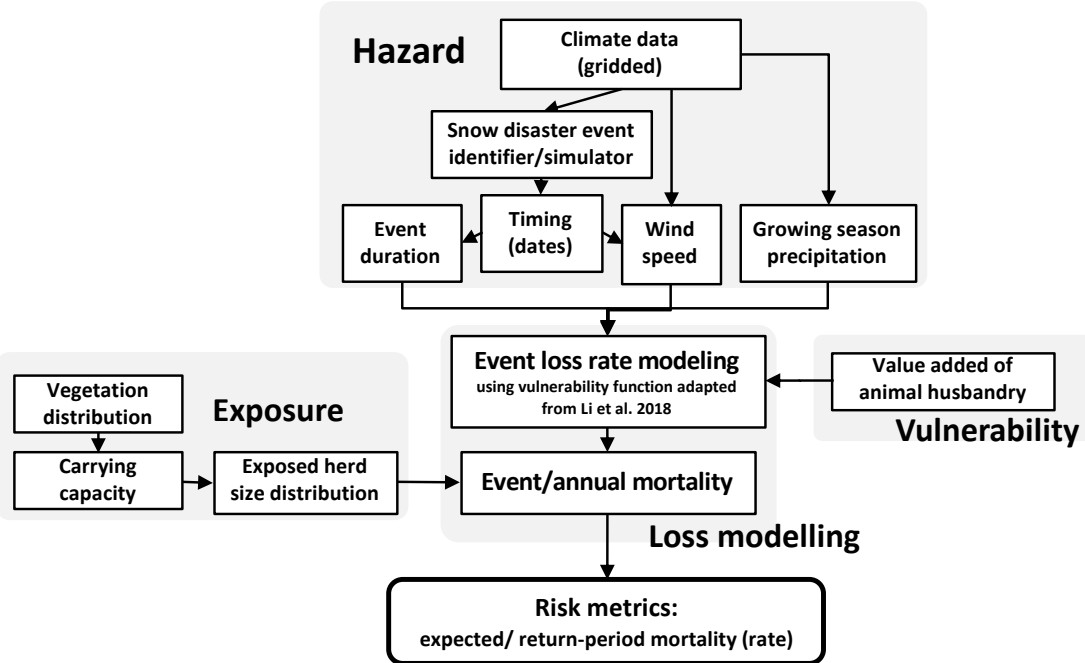

**Fig. 1 Event-based Probablistic Risk Assessment Framework for Livestock Snow Disasters in the Qinghai-Tibetan Plateau region**

### 2.2.1 Hazard

In our event-based PRA method, the Hazard module needs to identify individual snow disaster events to provide event duration (*Duration*) and during event wind speed (*Wind*), two important inputs to model event loss using the vulnerability function. It requires the exact timing (start and end dates) of each event. However, the timing of each individual event is not straightforward to obtain. For the historical period, there are no ready-to-use snow disaster event datasets at the grid level. The number of meteorological stations capable of observing snowfall in the QTP is limited and are primarily located in the eastern and

southern parts of the region. For future risk assessment, no projections of snow disaster events are provided in climate scenario datasets, although models have been developed to simulate daily snow depth (Yuan et al., 2016). Therefore, a snow disaster event identifier/simulator was developed here to identify and simulate snow disasters.

A snow disaster is a weather process with snow fall, low temperature, and snow cover, with certain length of durations, according to the Chinese national standard for *Snow Disaster Grades in Grazing Regions of China* (GB/T20482-2017) and China Meteorological Administration (CMA) standard for *Meteorological Grades of Urban Snow Hazards* (QX/T 178-2013). A snow disaster event designation largely depends on the snow weather process and observer's decision (manual record). To mimic a meteorological observer's decision to designate a snow disaster event, our snow disaster event identifier/simulator

has considered two major questions. First, whether a specific day would be regarded as a snow-disaster-day (SDD) given weather information of the day and previous days. The key is the modelling the binary response variables (Yes/No), which can be conducted with either regression or classification methods. Second, whether two SDDs, exactly neighbouring or a couple of days away from each other, should be regarded as one snow disaster event. The key is to assemble many single SDDs into snow disaster events, which can be accomplished using smoothing and filtering. In response, three major steps were considered

(Fig. 2, Fig. A1):

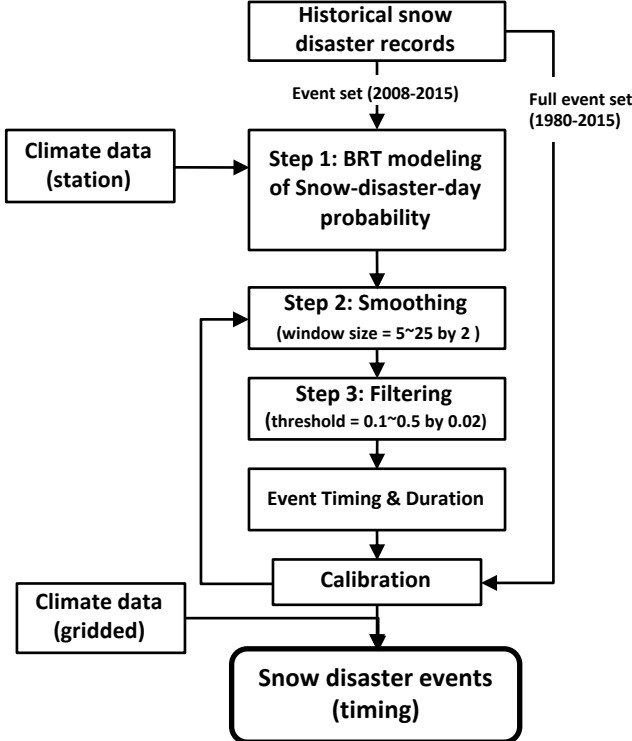

Fig. 2 Technical flow of the snow disaster event identifier/simulator

**(1) Step 1: Modelling SDD probability for each single day**

For this step, boosted regression tree (BRT) modelling was used to establish multi-variate and non-linear relationships between SDD and various weather information. The BRT modelling methodology was chosen due to its promising power for both explanatory and predictor purpose in many ecological and environmental modelling scenarios (Elith et al., 2008; Hastie et al., 2009). Other machine learning methods, such as random forest, can also be used but are less likely to outperform BRT according to the literature (Oppel et al., 2012; Youssef et al., 2016). To fit a BRT model, historical snow disasters were first turned into SDD flags: if a date was included in a historical snow disaster it was flagged with "1", and "0" otherwise. Variables used to explain and predict days that would be considered SDD were inspired by the two standards, GB/T20482-2017 and QX/T 178-2013. Both standards agree well for the important indicators that define a snow disaster. We included daily snow depth (*SD*, cm), daily maximum (*maxWind*), mean (*meanWind*) and minimum wind (*minWind*) speed (m/s), daily maximum (*maxT*), mean (*meanT*), and minimum (*minT*) temperature (°C), daily precipitation (*Pre,* mm), and average daily precipitation since the last snow fall day (precipitation > 0.1 mm) (*aveP*, mm/d). *aveP* was used to denote the diminishing impact/pressure from snowfall as time elapsed. Daily snow cover data were not considered as they were absent in future climate projections. Because our goal is to develop a model framework that can assess both present-day and future risk with climate projections, we refrained from using variables absent in future climate projections.

Historical snow disaster event data with the time of each event for each meteorological station were used to train the BRT model. These data were obtained from two sources. Records for 1980–2007 were a collection of snow disaster records published in six provincial meteorological yearbooks for the Tibetan Plateau (Wang et al., 2013b). Records from 2008–2015 were obtained from the China Meteorological Science Data Sharing Service System (CMSDS, http://data.cma.gov.cn). Records in both datasets are official releases of snow disaster records by meteorological administrations and are consistent with each other in terms of observation standards. Data for the predictors were also obtained from CMSDS, including 106 national reference stations in the region. The dataset contains daily observations of maximum, mean, and minimum temperature, maximum and mean wind speed, and precipitation.

BRT model fitting was conducted using the package *dismo* (Hijmans et al., 2011) in R 3.3.3. Given the type of response variable, the Bernoulli distribution family was used. The BRT model has two important required parameters to obtain the optimal prediction, tree complexity (*tc*), which is the number of splits in each tree and controls whether interactions are fitted, and learning rate (*lr*), which determines the contribution of each tree to the growing model (Elith et al., 2008). To identify the best combination of model parameters, we compared the combinations of *lr* = (0.01, 0.005, 0.001) and *tc* = (1, 2, 3, 5), as recommended by Anderson et al (2016). The maximum number of trees was set to 20,000, which proved sufficient for convergence. For each combination of parameters, we applied the predictor selection process using the *gbm.simplify* function, which uses a process of variable selection analogous to backward select in regression. It drops the least important predictor, then re-fits the model and sequentially repeats the process (Elith et al., 2008) until some stopping criteria, i.e., the reduction in predictive performance exceeds some threshold (Elith et al., 2008, Appendix S2). We used a 10-fold cross-validation and the result with the least cross-validation deviance was retained. To achieve the most promising goodness-of-fit, historical snow

disaster records obtained from CMSDS (2008–2015) were used for fitting. Records for 1980–2007 were used later for validation and calibration purposes.

**(2) Steps 2 and 3: assembling single SDDs to events by smoothing and filtering**

A fitted BRT model can help predict the probability of a single day being judged as a SDD. To predict/rebuild snow disaster events, these single day probabilities must be deemed snow disaster events, an ensemble of multiple SDDs. Because the explicit output from the BRT suffered from prediction errors, simply using a threshold to turn probabilities into "0/1" values would yield a set of "busy" snow disaster events, e.g., high frequency but small duration (Fig. A1, "Step 1"). Therefore, a smoothing treatment is needed to filter out isolated single SDDs and fill the small gaps between two neighbouring events. There are two parameters essential to changing the frequency and duration of identified snow disaster events: the smoothing window size and filtering threshold. In general, using larger window size for smoothing can filter out noises and reduce the frequency of events, while using lower threshold can increase the duration of single events. In order to best match the annual occurrence and the duration of single events, the two parameters were tuned through calibration using the full dataset of historical records between 1980 to 2015. We considered moving averages with window sizes from 5 d (minimum duration of a single disaster as defined by CMA) to 31 d (one month) in steps of 2 days, in combination with thresholds of 0.10 – 0.5 in steps of 0.02. The timing and duration of events derived from our model for any given pairs of window size and threshold were compared with historical records, including the frequency distribution of annual occurrence of single events, the frequency distribution of single event duration, and the timing of each single event. Through tuning, the combination of parameters that yielded the best matches were recorded.

Finally, the fitted BRT model together with the tuned parameters of smoothing and filtering was applied to generate all snow disaster events during 1980–2015 by grid. The China meteorological forcing dataset (He and Yang, 2011) obtained from the Scientific Data Centre of Cold and Arid Regions http://westdc.westgis.ac.cn/data/7a35329c-c53f-4267-aa07-e0037d913a21 was used. It offers variables, including precipitation, air temperature, wind speed, and sunshine duration at spatial resolution of 0.1° × 0.1° and temporal resolution of 3 h. We used this dataset because it focuses on the cold and arid regions in western China, and the QTP has been used as a focus region for validation (Chen et al., 2011; Yang et al., 2010). The 3-h dataset was aggregated to daily for input to the BRT model to rebuild gridded snow disaster events. Based on the identified events, the variables *Duration* and *Wind* were computed as inputs to the vulnerability function. From the 35 winters' events identified, we calculated the annual frequency and mean (single) event duration of snow disasters, as well as their return period values (Fig. A3).

### 2.2.2 Vulnerability function

Vulnerability is a critical function that links dose (hazard inputs) and response (loss estimates) (Carleton and Hsiang, 2016). For livestock snow disasters, a set of vulnerability functions have been estimated linking livestock mortality (rate) to snow disaster duration, within-disaster environmental stress, summer season vegetation productivity, and disaster prevention capacity (Fang et al., 2016; Wang et al., 2016). To fulfil the goal of event-based modelling, the vulnerability relationship must be built on an event basis. Using generalized additive models, Li et al. (2018) derived the quantitative relationship between

livestock mortality (rate), snow disaster event duration, within-disaster wind speed, pre-winter vegetation condition, and time index. Using the identical dataset, we included disaster prevention capacity in the analysis, using socioeconomic indicators as a proxy, and followed Li et al. (2018)'s approach to derive the model with best predictive power. We tried different socioeconomic indicators, including gross domestic production, value added of animal husbandry, fiscal revenue, fiscal

expenditure, and gross domestic production per capita, following suggestions from the literature (Wei et al., 2017). We found the model using value added from animal husbandry yielded the best fitting result, having a deviance-based $R^2$ of 0.625; more details of the model, including model fit statistics, response curves and model performance diagnostics, are provided in supplementary material S1. Therefore, the following version of the generalized additive model was considered in further analysis:

$$\ln LR = s(Duration) + s(Wind) + s(P) + s(Value\_Add) \,, \tag{1}$$

where, the livestock mortality rate induced by a snow disaster is determined by disaster duration (*Duration*), within-disaster maximum daily mean wind speed (*Wind*), growing season (May–Sep) aggregate precipitation (*P*), and prevention capacity as measured by value added of animal husbandry (*Value_Add*) of the underlying county. *Duration* was used as the key indicator of hazard intensity. *Wind* and *P* were used to denote within-disaster and pre-season environmental stressors, respectively (Li

et al., 2018). *Value_Add* was used to indicate disaster prevention capability, which explicitly measures the size of animal husbandry, and implicitly represents prevention infrastructure and capability of risk management (Wei et al., 2017).

Given such a relationship, the vulnerability is a truly dose-response function between livestock mortality rate (mortality/herd size) and snow hazard intensity together with other environmental stressors and prevention capacity, as proposed by (Carleton and Hsiang, 2016). Different from simply defining vulnerability as the loss rate (Jongman et al., 2015; Kinoshita et al., 2018),

the potential influence from socioeconomic development is embedded in the vulnerability function.

### 2.2.3 Exposure

Exposure measures the distribution of assets or population exposed to hazards (Kinoshita et al., 2018). In our framework, it provides the spatial distribution of herd size/density exposed to snow disaster and converts outputs from event loss modelling and livestock mortality rate (the response variable in the modelled vulnerability function) into mortality (death toll). By

definition (Fernández-Giménez et al., 2012), livestock in nomadic grazing are most prone-to snow disaster because they obtain food mostly from grassland. Livestock raised in ranches or industrial livestock farms in agricultural regions, by contrast, are much less exposed because they have steady food supplies from crop by-products and are well protected by infrastructure. Therefore, the estimated number of livestock grazing on grassland were used to denote livestock exposure to snow disaster.

A full gridded distribution map of herd size/density grazing on grassland in the QTP is not directly available, but it can be

derived according to the rule-of-thumb for "forage-livestock balance". According to the *Forage-livestock Balance Management Approach,* issued by the Ministry of Agriculture of China in 2006 to mitigate severe over-grazing in the pastoral areas of China (Shang et al., 2012), herd size grazing on grassland at the county level must be strictly controlled under carrying

capacity. Therefore, a gridded carrying capacity map is a good approximation of actual herd size/density distribution exposed to snow disaster.

There are several factors that determine the carrying capacity of a given region, but the most important is grassland type, according the *Ministry Standard of Calculation of Rangeland Carrying Capacity* issued by Ministry of Agriculture of China (NY/T 635-2015). In the standard, grassland type was used as the key identifier for determining forage regrowth percentage, proper utilization rate of rangeland in different grazing seasons, and the conversion coefficient for standard hay. Therefore, we estimated the spatial distribution of exposure (sheep unit/ha) from grassland distribution data using a look-up table for grassland-type to carrying-capacity relationship. For the look-up table, we adapted the plan of Xin et al. (2011) for Qinghai. For Tibet, we reviewed various criteria (Zhang et al., 2014) and followed the official release of the Autonomous Region government (Land Management Administration of Tibet Autonomous Region, 1994; Department of Agricultural and Pastoral of Tibet Autonomous Region, 2011). The final look-up table is supplied in Appendix (Table A1). For grassland distribution, we used the Vegetation Map of the People's Republic of China (1:1 million) (Editorial Committee of Vegetation Map of China and Chinese Academy of Science, 2007), which offers detailed information about the spatial distribution of 11 vegetation type groups, 55 vegetation types, 960 plant formations, and more than 2000 dominant species in vector data. To match the look-up table and map information, we merged some vegetation types and used only the major grassland types (percentage area >0.5%) according to a survey from the Food and Agriculture Organization of the United Nations (FAO, 2005) (Fig. A2; Table A1). The estimated carrying capacity was aggregated to the county-level and compared to the official release of Tibet Autonomous Region. The two datasets showed good agreement, with a correlation coefficient of 0.769. Therefore, the estimated carrying capacity was used as exposure to convert mortaility rate into mortality.

**2.2.4 Loss modelling**

Snow disaster event losses measured with livestock mortality rate (death toll/ herd size) were modelled by taking requested inputs into the vulnerability function. *Duration* and *Wind* were outputs from the hazard module. Growing season aggregate precipitation *P* was computed from the climate forcing data. Year-end county-level *Value_Add* were obtained from the statistical year books of Qinghai, Tibet, Sichuan, Gansu, and Xinjiang. County-level *Value_Add* values were assigned to each grid within its boundary. When modelling losses, we considered two cases:

1) Loss based on historical prevention capacity. To modell actual historical loss for model calibration and validation purposes, we used the actual county-level *Value_Add* of the study area for 1980–2015. Because *Value_Add* increases with time, it indicates the increasing prevention capacity, and therefore declining livestock mortality (rate) with time.

2) Loss based on present prevention capacity. For risk assessment purposes, we used the constant value of *Value_Add* from 2015 for two reasons. First, we needed to fit probability distributions over the modelled loss to derive final risk metrics, and the process required that the underlying loss samples were least stationary in their means and variances. Using a constant *Value_Add* value for 2015 avoided introducing trends inherent within *Value_Add* into the modelled loss, as the *Value_Add* has been growing. Second, because we assumed that *Value_Add* is a proxy for prevention capacity, using *Value_Add* value for 2015 in loss modelling helped estimate the potential loss given very recent prevention capacity (year

2015) rather than those of the 1980s or 1990s. Therefore, the derived risk metrics are more helpful for prevention planning and insurance implications in the near future.

The searching of snow disaster event and modelling of loss starts in every August and ends in June of the next year. Event mortality rates were then aggregated into annual mortality rates, considering the possibility of multiple events per location annually, although unlikely. In aggregation, we assumed that the second snow disaster event can only have an impact on livestock surviving from the first event, and so on. Therefore, the annual aggregate loss rate in a given grid is

$$\Delta = 1 - \prod_{i=1}^{N}\left(1 - \delta_i\right),$$ $i = 1, 2, \ldots, N$, in which $\delta_i$ is the modelled loss rate of the $i$th event in a year, and $N$ is the total number

of events. The annual aggregate mortality rate can finally be turned into a death toll by multiplying exposure, the herd size in a given grid.

Event/annual mortality (death toll) can then be derived by multiplying event/annual loss rate in any given location by its herd size. For each grid, 35 annual loss records were modelled (there are 35 winters in 36 years), including both mortality and mortality rate figures. The number of event loss records differ by location, depending on the identified number of events for each grid.

### 2.2.5 Risk metrics

In the risk metrics, modelled losses of discrete event/annual losses were turned into a probability distribution of losses. We followed standard risk metrics by deriving the average and return period values (Michel-Kerjan et al., 2013; Shi and Kasperson, 2015) of annual mortality rate and death toll for each grid. Model-derived annual mortality rates, based on constant *Value_Add* for 2015, were used to derive risk metrics. Due to our limited time span for repetition, return periods of 10 years (the 90th percentile of the distribution), 20 years, and 50 years were considered, while the 100-year usually used in flood/ earthquake studies (Kinoshita et al., 2018) was not considered. The kernel density method was employed to fit non-parametric distributions to derive the return period values by grid. We used the Gaussian kernel function and its corresponding optimal window width in the fitting process according to the "rule-of-thumb" for optimality (Deng et al., 2007; Silverman, 1986). In addition, aggregate mortality rate and death tolls at the municipal level were derived using zonal statistics to better validate the result with historical losses, and provide policy implications.

### 3. Results

### 3.1 Modelled snow disaster duration, frequency, and annual loss

### 3.1.1 The trained BRT model and tuned parameters in rebuilding snow disaster events

The trained BRT model retained six variables but excluded *SD*, *minWind*, and *Pre* as the result of the predictor selection process. In the final model, we used $lr = 0.001$ and $tc = 5$. It had a training data Area-Under-the-Curve (AUC) score of 0.948, and a cross-validation AUC of 0.909, indicating good prediction performance (Youssef et al., 2016). For the six variables

entered in the final model (**Fig. 3**), *maxT* had the highest relative contribution[2] (32.77%), while *meanWind* had the lowest (5.78%).

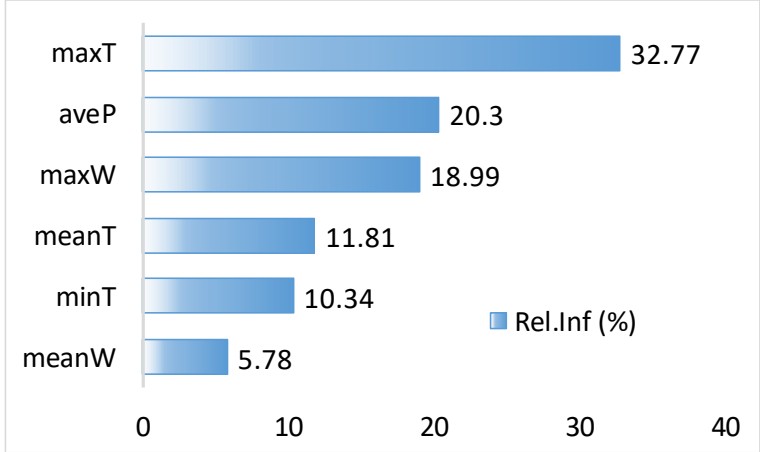

**Fig. 3 Relative influence of variables predicting a snow-disaster-day. Blue bars are relative importance of each factor and the sum of all relative importance fractions is 100%.**

After tuning the window size with the moving average and threshold, we found the best results with a window size of 21 and threshold of 0.18. The derived results captured the timing of occurrence of historical events (**Fig. A1**) and matched the empirical cumulative density functions (ECDF) for historical durations (**Fig. 4**), for both event and annual aggregate durations. In historical records, two or more events in a single year at a single location are rare. Therefore, ECDFs for historical single event duration and annual aggregation duration were quite close to each other in Fig. 3. Two-sample Kolmogorov–Smirnov tests were also conducted to verify the degree of agreement between ECDFs. For single event duration (observed vs. predicted), the test statistic was 0.138, and its corresponding p-value was 0.118. The annual aggregate duration (observed vs. predicted) test statistic was 0.131, and its corresponding p-value was 0.189. Therefore, the prediction model performed well in capturing the statistical features of historical snow disaster duration and the predicted results can be used for event loss modelling.

---

[2] In BRT, the relative importance is calculated based on the number of times a variable is selected for splitting, weighted by the squared improvement to the model as a result of each split, and averaged over all trees. The relative importance for each variable is scaled so that the sum adds to 100 (Elith et al., 2008).

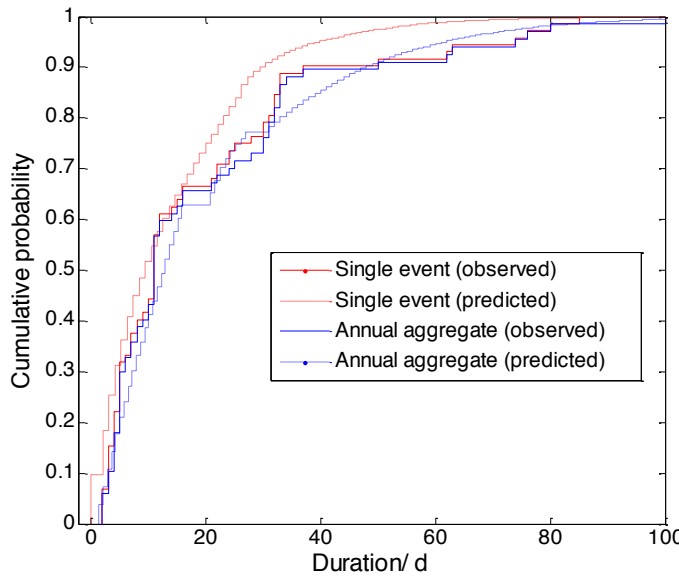

**Fig. 4 Empirical cumulative density functions for historical and model-predicted snow disaster duration**

### 3.1.2 Model-derived snow disaster events, 1980–2015

With the tuned model, the timing of snow disaster events were identified in the historical period 1980–2015. Correspondingly,
the annual occurrence frequency and duration of snow disaster events were derived (Fig. 5). In the figure, non-grassland areas, including permanent snow areas, were masked using the vegetation map. Across the entire plateau, the annual average frequency was below 0.2 in most regions, i.e., on average, snow disasters occur every 5 years in these regions. Higher frequency regions were primarily located in major mountains, including the Tanggula Range and Nyainqêntanglha Range in the central part of the plateau, and the Kailas Range and neighbouring Himalayas. These regions are higher elevation and spatially close
to permanent snow-covered areas. For major pastoral production regions, i.e., the Naqu prefecture in the central QTP, the annual average frequency was 0.2 to 1, echoing the local proverb, "small disaster once in 3 years, and a major disaster once in 5 years" (Ye et al., 2017b).

The distribution of mean annual aggregate duration of snow disasters was consistent with annual frequency, indicating strong controls from elevation and topology. For most regions, mean annual aggregate duration was below 3 d. For typical pastoral
regions, i.e., Naqu, a snow disaster can last for more than 14 d on average. In comparison, the annual aggregate duration can last for more than 21 d in high elevation mountainous areas, including the Himalayas in the southwest and alpine meadows to the east end of the Bayankhar Mountains, which is nearly 10% of the total grids with valid values.

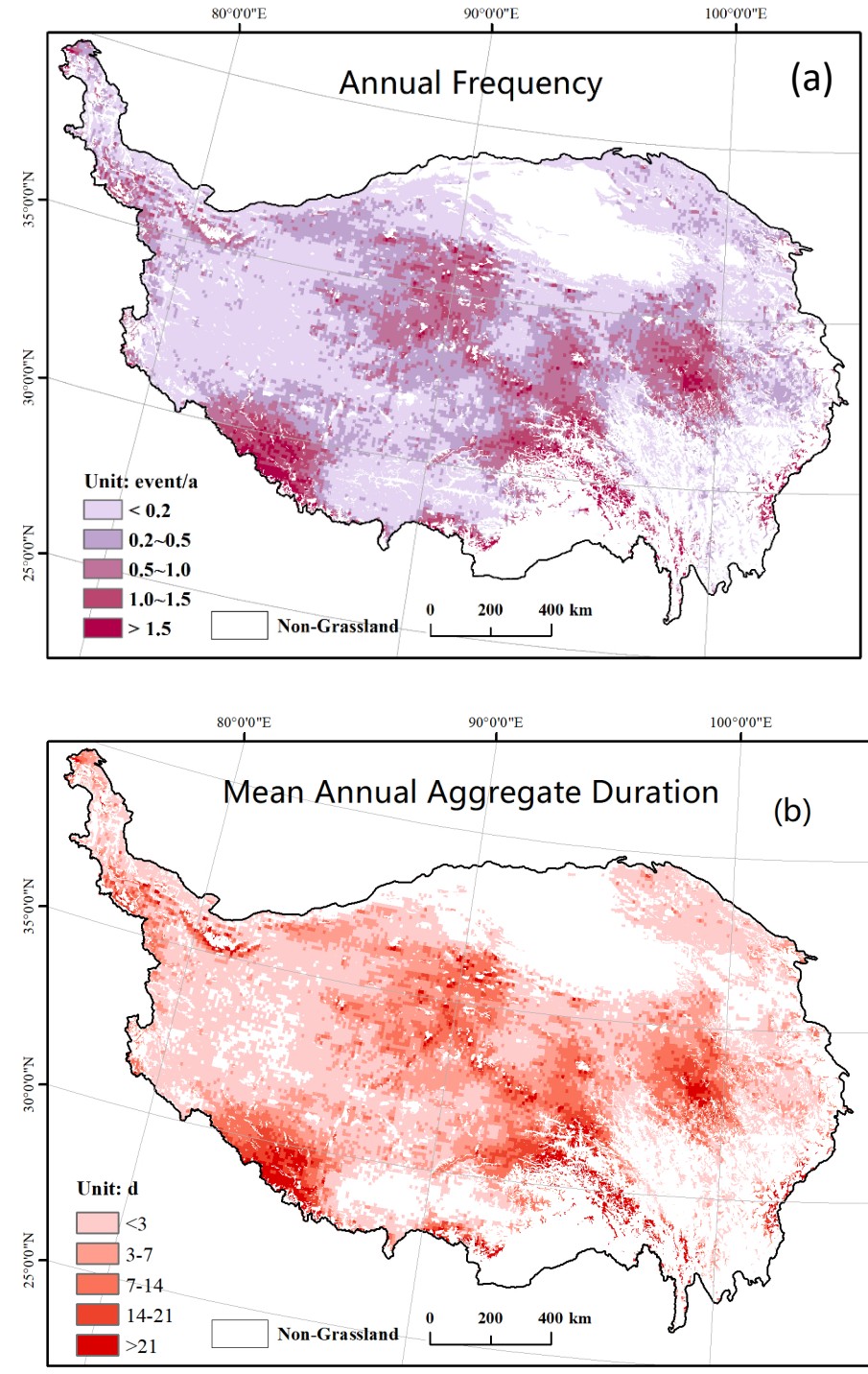

**Fig. 5 Gridded annual frequency/ annual average occurrence (a) and mean annual aggregate duration (b) of snow disasters from model predictions**

### 3.1.2 Model-derived annual snow disaster loss, 1980–2015

Model-derived annual snow disaster losses (1980–2015) are provided in Fig. 6. The orange time series show losses modelled using *Value_Add* from historical values (dynamic), assuming historical prevention capacity. The blue time series show losses modelled using constant *Value_Add* from 2015, assuming present-day prevention capacity. All losses are for a specific snow disaster season from August to the next June, rather than a civil year.

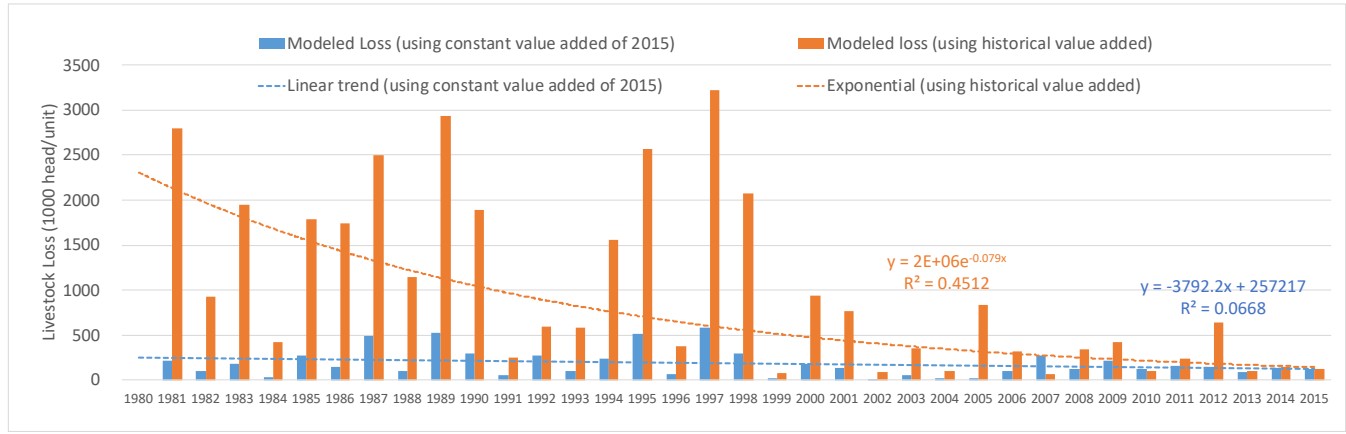

**Fig. 6 Model-derived annual livestock loss from snow disasters in the QTP (1980–2015).** The unit of modelled loss has been converted from sheep units to heads/units by dividing by 2.2 (total sheep units ≈ 2.2×herd size by heads/units according to the cattle/sheep structure in the QTP).

To measure model performance, historical losses over the QTP for 1980–2015 were collected from the China Meteorological Disaster Catalogic (Wen 2008) (for 1980–2000) and China Meteorological Disaster Yearbook (2004–2016) (China Meteorological Administration, 2004–2016). The model result did capture the interannual variation of losses: the correlation coefficient of the modelled loss and recorded historical loss was 0.688 and the root-mean-square-error was 250,841. Our model also captured most years that experienced severe snow disaster loss, i.e., major loss years with annual aggregate loss of over 500,000 heads/units. These years include 1981 (referring to 1981 snow season, August to June of the next year), 1982, 1985, 1986, 1988, 1989, 1992–1995, 1997, 2007, and 2012. The correlation coefficient of the modelled loss and recorded historical loss of these years was 0.779, the root-mean-square-error was 400671, and mean-absolute-percentage-error was 37%. For the peak loss years (annual aggregate > 2 million), model results were also good. The modelled vs. recorded losses were 2.79 and 2.48 million for year 1981, 2.07 and 2.93 million for 1989, 2.94 and 2.57 million for 1995, and 2.79 and 3.22 million for 1997.

The modelled historical loss also exhibited a clear decreasing trend compared to the modelled loss associated with present day prevention capacity (the blue series). The difference indicates that an improved prevention capacity, using value added of animal husbandry as a proxy, played important role in reducing livestock loss in snow disasters. In addition, it also confirmed that the modelled historical loss cannot be used for directly fitting the probability distribution of loss due to its pronounced trend. Instead, the modelled loss associated with present day prevention capacity is appropriate.

## 3.2 Probabilistic risk assessment results

### 3.2.1 Risk in terms of livestock mortality rate

The assessed livestock snow disaster risk measured using the annual mortality rate is presented in Fig. 7. Because it is not viable to present the full probability distribution of livestock mortality rate by grid, these figures include the annual average and three return-period mortality rate maps (10-year, 20-year, and 50-year), upon which the non-pasture areas were masked. Spatial distributions of mortality rate for different return-periods are highly consistent (Fig. 7). The pattern is very similar to the pattern of annual aggregate snow disaster duration (Fig. 5), confirming the dominant influence of snow disaster duration. High-mortality rate regions are primarily located in the major mountainous areas, including the Tanggula Range and Nyainqêntanglha Range in the central QTP, the Kailas Range and neighboring Himalayas in the southwest QTP, Bayankhar mountains in the east QTP, and southern part of the Kalakoram Range and west-end of the Kunlun Mountains in the northwest corner of the QTP. Classified by administrative districts, high mortality rate regions include the Yushu and Guoluo Prefectures in Qinghai Province and Naqu, southwest Ngari, and Northwest Shigatse Prefectures in the Tibet Autonomous Region. In these regions, the annual average mortality rate reaches 10% and in some parts of Guoluo and Shigatse, the 50-year mortality rate can reach more than 10%.

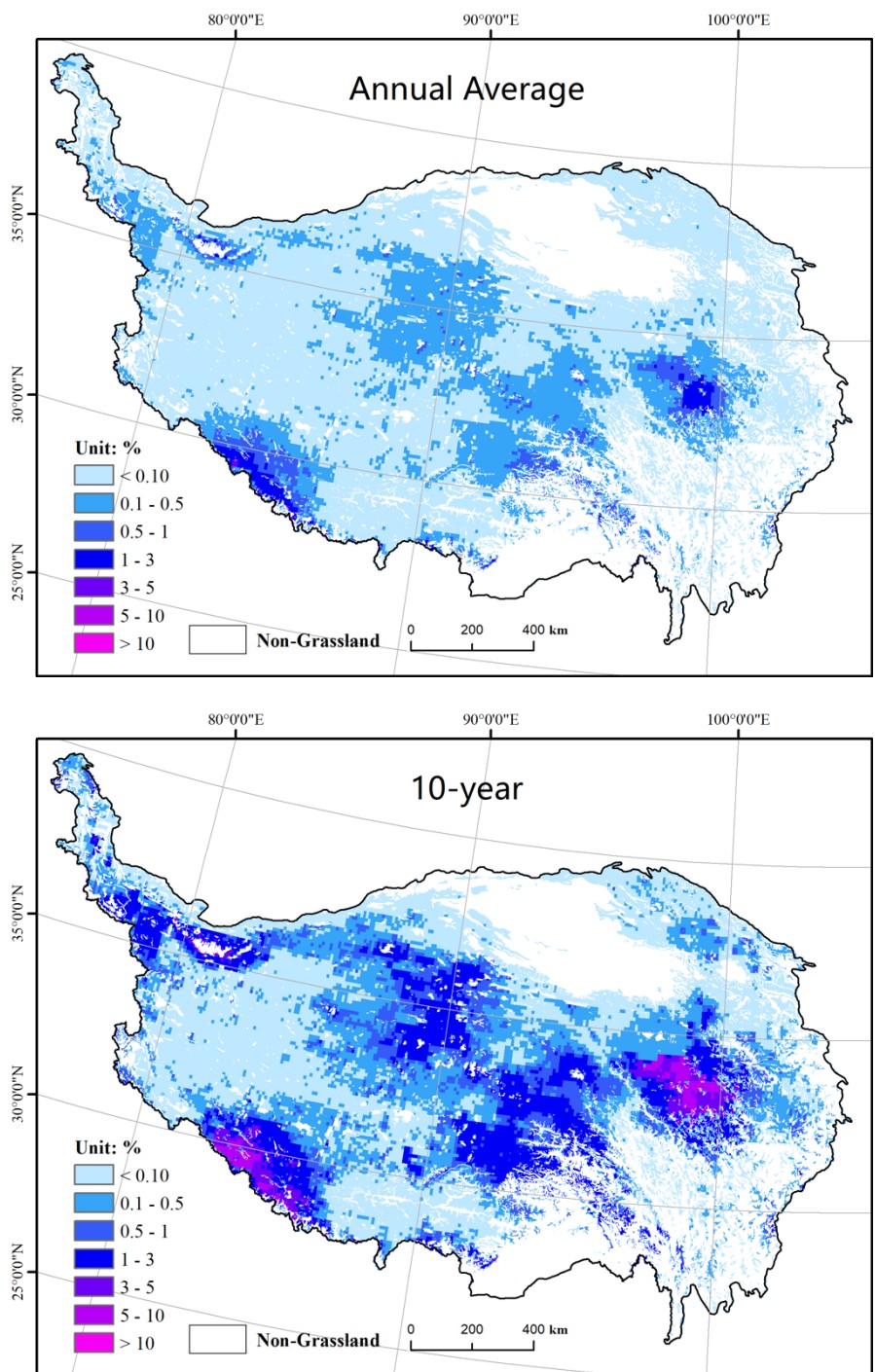

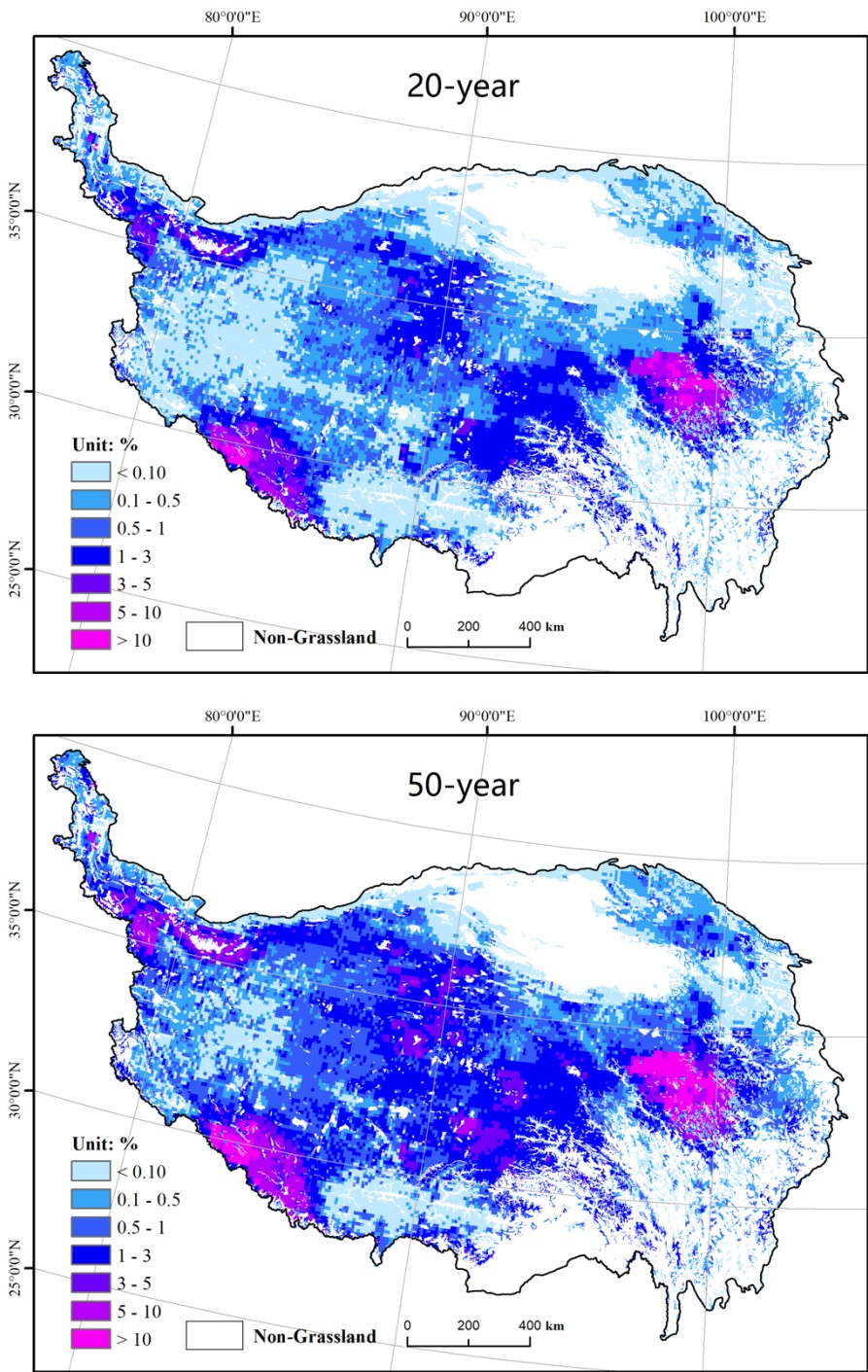

**Fig. 7 Gridded livestock snow disaster risk in terms of mortality rate (%) in annual average values, and 10-year, 20-year, and 50-yar return-period values. The grid size is 0.1° × 0.1°.**

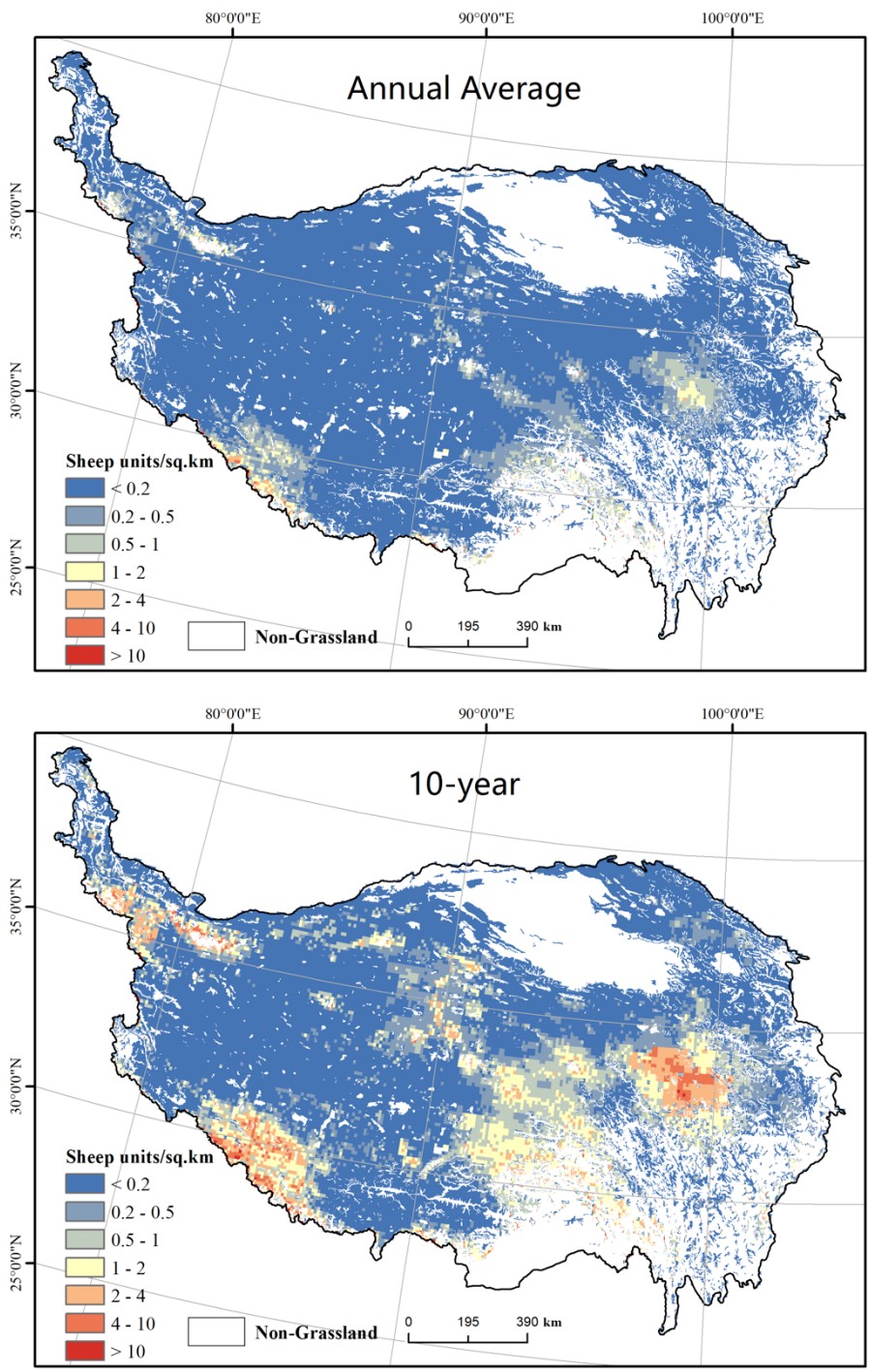

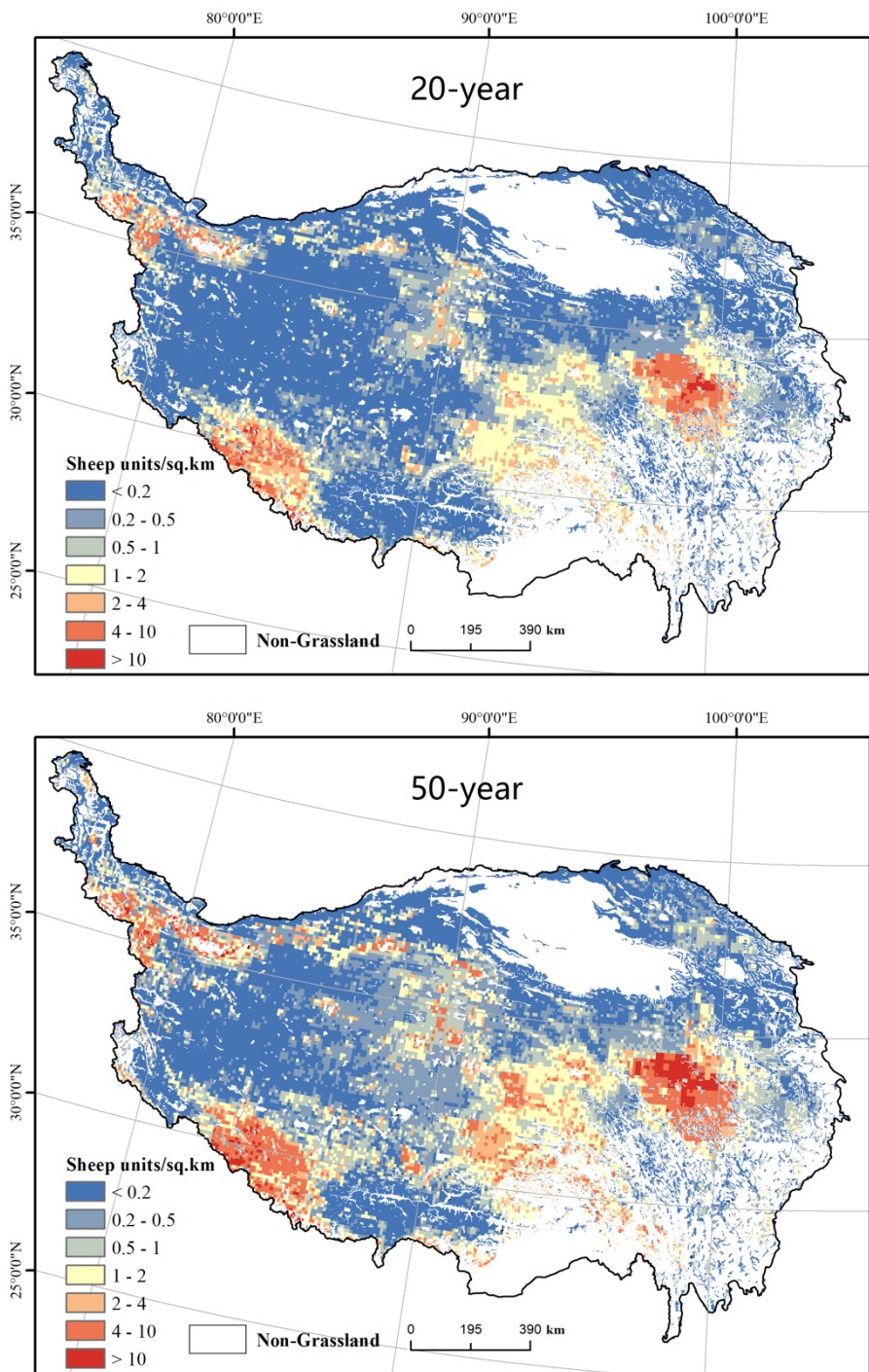

**Fig. 8 Gridded livestock snow disaster risk in terms of mortality (sheept units/km²) in annual average values, and 10-year, 20-year, and 50-yar return-period values.The grid size is 0.1° × 0.1°.**

### 3.2.2 Risk in terms of livestock mortality

Risk metrics in terms of livestock mortality were derived by multiplying the mortality rate by exposure (Figure 8). Again, annual average mortality, and the mortality at 10-year, 20-year, and 50-year return periods are all reported.

Mortality appears small in Figure 8, generally several sheep units/km$^2$. However, when aggregated at the prefecture level, mortality remained considerable (Table 1). Zonal statistics results identified high risk prefectures, including Guoluo and Yushu in Qinghai Province and Naqu, Shigatse, and Nagri in the Tibet Autonomous Region. In these prefectures, annual mortality with a return period of 20-year was mostly greater than 200,000 sheep units, which is the threshold for an extremely severe livestock snow disaster, as defined in GB/T20482-2017. Among them, Guoluo, Naqu, and Shigatse are of extremely high risk. Their 20-year mortalities were all >500,000 sheep units.

**Table 1 Livestock snow disaster risk in terms of mortality (1,000 sheep units) by prefecture**

| Prefecture | Herd size exposed (estimated) | Mortality (annual average) | Mortality (20-year) | Mortality (50-year) |
|---|---|---|---|---|
| Xining | 283.4 | 0.1 | 3.9 | 5.6 |
| Haidong | 437.2 | 1.3 | 6.3 | 13.1 |
| Haibei | 2036 | 1.0 | 26.5 | 37.9 |
| Huangnan | 1080.2 | 0.6 | 13.8 | 19.4 |
| Hainan | 1692.7 | 0.8 | 23.7 | 34.2 |
| Guoluo | 5160.7 | 84.9 | 1098.1 | 1962.5 |
| Yushu | 12720.9 | 28.8 | 492.1 | 771.2 |
| Haixi | 10659.2 | 6.7 | 129.0 | 187.3 |
| Lahsa | 1802.1 | 0.4 | 9.6 | 13.9 |
| Changdu | 4217.4 | 12.7 | 155.6 | 219.4 |
| Shannan | 2607.5 | 0.2 | 5.6 | 7.9 |
| Shigatse | 12245.4 | 110.9 | 932.4 | 1470.7 |
| Naqu | 17877.2 | 48.7 | 646.8 | 952.6 |
| Ngari | 12970.8 | 12.8 | 218.8 | 334.3 |
| Linzhi | 2743.9 | 9.4 | 120.2 | 166.9 |

Note: Only prefectures with a majority of land mass within the QTP are listed. Statistics reported in the table only refer to areas within the QTP.

# 4. Discussion

## 4.1 Spatial patterns of livestock snow disaster risk in the QTP

Our results illustrate the spatial distribution and offer quantitative metrics of risk in terms of livestock mortality and mortality rate due to snow disasters in the QTP. The spatial pattern of risk agrees with earlier studies covering this region quite well.

From an empirical perspective, the literature frequently mentions Easter Inner Mongolia, the Northern Tianshan Mountains in Xinjiang, and Northeastern QTP as centers of snow disaster around China (Gao, 2016; Hao et al., 2002). Within the QTP, high frequency snow disaster regions that are mentioned repeatedly in the literature include Yushu, Guoluo, Naqu, Shigates, and Nagri (Bai et al., 2011), which have all been identified in our study. As for risk assessment, our results also agree well with earlier studies. For instance, regions between the Kailas Range and neighboring Himalayas, southern Qinghai Province (mainly

Yushu and Guoluo), and the northwestern corner of the QTP are all considered as higher risk regions in both qualitative (Liu et al., 2014b) and quantitative (Shi, 2011; p.106–107) risk assessment results. In northern and western Naqu Prefecture, and the central-to-western end of the Nyainqêntanglha Range, our results are consistent with the national snow disaster risk map (Shi, 2011; hereafter termed risk maps), which are of higher-to-the-highest risk. Nevertheless, these regions are considered the lowest of lower risk in the results presented by Fenggui Liu et al. (2014).

Our results for the magnitude of annual average mortality rate were smaller than those in the risk maps of China (Shi, 2011; p.104–107); in general, our values were about half those previously reported. For the high-risk regions, annual average mortality rates were generally $\geq 2\%$ in our results, but $\geq 4\%$ in the risk map results. Our result had a vast low risk region with annual average mortality rates <0.5%, but the threshold was 1~3% in the risk maps. In terms of mortality, our results matched historical records better. For instance, the most severe and deadliest snow disaster in southern Qinghai Province since 1960

was in the 1995–1996 snow season, mostly in Guoluo and Yushu (Bai et al., 2011). It claimed a loss of 1.20 million livestock (Wen, 2008). According to our model-derived historical losses for Guoluo and Yushu Prefectures, the mortality was approximately 1.20 million heads/units in 1996 (converted from 2.64 million sheep units modelled given the cattle-to-sheep ratio in Qinghai Province). Another example is the 1997–1998 snow disaster in Naqu, the most severe snow disaster since 1960, leading to a loss of $0.82 \times 10^6$ livestock (Wen, 2008). Our model-derived mortality for this event was 0.72 million

head/unit (turned from 1.59 million sheep units). The 1995–1996 Yushu and Guoluo snow disaster would have a return period over 50-year, and the 1997–1998 Naqu snow disaster would have a return period of 80 to 100-year according to our metrics (Table 1). If the mortality rate estimated in risk maps were used instead, then the corresponding return-period could be underestimated, and consequently, snow disaster risk exacerbated.

## 4.2 Temporal changes in livestock snow disaster loss and its drivers

Our results rebuilt a complete list of annual livestock snow disaster losses for the 1980–2015 period (Fig. 5). The modelled loss shows a clear declining trend. Major and peak loss years occurred frequently before year 2000, but rarely after that. Using a different historical data set, Wei et al. (2017) suggested that based on trends from 1960–2015, the loss would increase in the

long run. However, focusing on the later part of their dataset, i.e., 1980–2015, similar downward trending results would have been derived.

Our results indicate that both climate change and improved prevention capacity have contributed to the declining trend in annual livestock loss. The effect of climate change is revealed by the model-derived historical loss using a constant value added from animal husbandry (the blue series in Fig. 6). It has a very modest declining trend, -3792 head/unit per year, or equivalently -1.8% per year if an exponential trend was applied. Such a modest declining trend after controlling for prevention capacity is supported by the literature. Earlier studies reported a uniform increase in temperature (Kuang and Jiao, 2016), reduced snow cover area (Duo et al., 2014), snow depth and snow cover days (You et al., 2011), and increased growing precipitation and improved vegetation (Pang et al., 2017) in the QTP. All of these factors contribute to smaller event frequency, shorter duration, and less environmental stress during and before the snow season.

Improved prevention capacity plays a much more significant role in declining annual livestock losses. This conclusion is supported by the difference between the two model-derived annual loss series in Fig. 6 because they share an identical historical snow disaster event set and differ only in prevention capacity. For model-derived historical losses (orange series in Fig. 6), the exponential trend indicates that annual livestock losses decrease 7.9% per year, or 57349 head/unit per year if a linear trend is applied. Therefore, improving prevention capacity accounted for an approximately 6.1% reduction in annual livestock loss per year if exponential trends are assumed, or – 53557/a if linear trends are assumed. This contribution is supported by reported government investments in infrastructure, such as thermal barns/sheds and fenced grassland, and improved preparedness, such as winter season hay and forage storage, although there remains significant potential for further improvements (Shang et al., 2012; Wang et al., 2013b; Wei et al., 2017).

## 4.3 Advantages of the event-based PRA

Our study differs from the existing literature largely in its event-based PRA framework. Such a framework derives unique information, which were not obtained in earlier methods based on annual analyses and are important for preparedness decisions and insurance solutions. With the event-based PRA framework, the following information is derived for better risk reduction. 1) The event-based framework provides an estimate of the frequency distribution of single disaster events. Overall, our analysis indicates that snow disasters are frequent in terms of annual occurrence, but more than one snow disaster a year is unlikely (Fig. 5). Given this finding, counter-measures can be implemented to build prevention capacity to handle one event annually (Mechler et al., 2010). In addition, the framework can be further applied to climate change analysis. Our snow disaster event identifier can help reveal the changes in frequency and intensity (mainly *Duration*) of snow disasters in response to climate change, and therefore provide information for adaptation.

2) Our results for single event duration provide important quantitative references for hay and fodder storage, which were not achieved by earlier annual basis analyses. For the majority of higher-risk regions, once a snow disaster occurred, it lasted for an average of 12 d (Fig. 5). At return periods of 10-year and 20-year, the durations of single events were up to 21 days and 28 days, respectively (Fig. A2). At return periods of 50-year, single events could last for more than 40–50 days. The regional average duration of a 20-year event in Naqu, Yushu, Guoluo, and south Ngari, was estimated to be 24, 22, 26, and 26 days,

respectively. From a preparedness perspective, the amount of hay and fodder storage needed from herder households and local government reserves combined, can be readily estimated from our results once their goal of preparedness capacity is set, i.e., capable of managing a 10-year event. Alternatively, our results can also help local regions measure their preparedness capacity given their amount of hay and fodder storage. For instance, according to the authors' survey (Ye et al., 2017b), the total amount of hay purchased can only support supplementary feeding of county-wide livestock for at most 3~5 days in some counties in central Naqu Prefecture. Such a level of preparedness can only endure a snow disaster with a less than 5-year return period.

3) Our event-based PRA results can also provide solid technical support for insurance solutions. Earlier studies that assessed risk on an annual basis using annual aggregate snow-cover days, or snow depth variables were incapable of doing so because insurance indemnities are clearly triggered by specific events. The frequency distribution of event occurrence and event duration provides necessary information to help the design insurance trigger schemes. These insurance products can be conventional (indemnity-based), where the post-disaster loss-adjustment is conducted based on herder households. In addition, our results can readily support calculating actuarially fair premium rates and at-risk loadings by applying deductible conditions, which can turn event loss records into event-based insurance losses (Wang and Zhang, 2003; Ye et al., 2017a).

## 4.4 Limitations

Several limitations in our risk assessment model must be mentioned. First, our hazard module to rebuild/predict snow disaster still suffers from uncertainty. We obtained a good AUC score from the BRT model for identifying snow disaster days, and also good agreement in timing of occurrence and distribution of duration for longer-duration events. However, the performance in capturing small disasters of short-duration, i.e., < 5 d, still needs improvement.

Because the exact spatial distribution of sheep units is unavailable, exposure data were derived according to the computed carrying capacity by grassland type. The total herd sizes computed differ within 20% of those officially released. In addition, for the historical period, prior to the implementation of the forage-livestock balance policy, the actual herdsize exposed would be larger than carrying capacity due to over-grazing. Therefore, because our estimated herd size was smaller than the potentially larger grazing herd, our model-derived historical loss was conservative. For risk assessment purposes, using present day exposure is reasonable to estimate livestock loss distribution in the next couple of years. For short and mid-range future risk assessment, a projection of exposure will be needed, which will require projected future grassland structure and productivity changes (Gao et al., 2016).

Finally, our risk metrics were derived from events rebuilt from historical climate data, but not from stochastic simulations. Consequently, we have a limited number of events and annual loss records. We are only confident in risk metrics less than 1/35 a. Metrics for any higher return periods were derived from extrapolation and must be used with caution. This limitation can be resolved by inputting stochastic climate datasets using a stochastic weather simulator.

## 5. Conclusions

Quantitative risk metrics derived under a probabilistic risk assessment framework are critical for understanding disaster risks and providing quantitative evidence for risk-informed decision-making and resilience-building. In this study, we developed an event-based PRA approach for livestock snow disaster in the QTP region and derived risk metrics for livestock mortality and mortality rate. Our assessment results show that the spatial distributions for mortality rate and mortality size are quite similar. Hazard intensity, in terms of disaster duration, was the major driver of spatial differences in livestock mortality, while the influence from exposure in terms of herd size was quite modest. High risk regions include the Nyainqêntanglha Range, Tanggula Range, Bayankhar mountains, and the region between the Kailas Range and neighboring Himalayas. At a return period of 20-year, the annual livestock mortality rate was estimated to be > 2% and mortality was estimated to be > 2 sheep unit/km$^2$. At prefecture levels, the most important animal husbandry bases were identified as high risk regions, including Guoluo in Qinghai Province and Naqu, and Shigatse in the Tibet Autonomous Region. In these prefectures, a snow disaster event with return period of 20-year a or higher can easily claim a total loss of more than 500,000 sheep units. Our results of return-period mortality rate and death toll show better agreement with historical losses than those reported earlier.

Compared to earlier results, our approach relies on the prediction/simulation of snow disaster events, and correspondingly the modelled livestock losses are on event basis. In addition, our quantitative results for the return-period disaster duration are valuable for preparing hay and fodder reserves and designing insurance protection. The methodology developed here can be further adapted to future climate change risk analysis and providing risk-informed adaption suggestions for the QTP region.

## Acknowledgements

This study was supported by the National Key R&D Program of China (grant number 2016YFA0602404), Fund for Creative Research Groups of the National Natural Science Foundation of China (grant number 41621061), and State Key Laboratory of Earth Surface Processes and Resource Ecology. We also thank two anonymous referees and the editor for their comments and suggestions on an earlier draft.

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

**Appendix**

**List of Abbreviations**

| | | |
|---|---|---|
| | AUC | Area-Under-the-Curve |
| 5 | BRT | boosted regression tree |
| | CMA | China Meteorological Administration |
| | CMSDS | China Meteorological Science Data Sharing Service System |
| | PRA | probabilistic risk assessment |
| | QTP | Qinghai-Tibetan Plateau |
| 10 | SDD | snow-disaster-day |

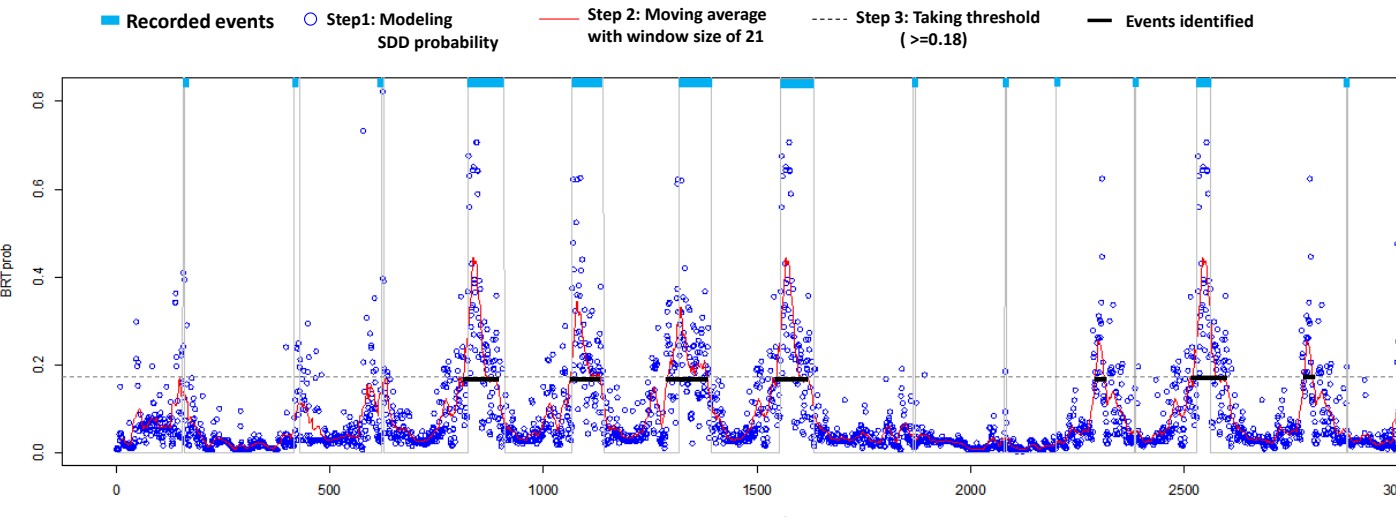

**Fig. A1 Illustrative chart of the procedure for identifying snow disaster events, and its calibration with historical records.** The time series is
of days in the winter season (October 1 to May 31) with snow disaster records after 2008. In total, this includes 13 station•winter and 3168 single c
shows that the procedure is capability of accurately capturing major historical events with relatively longer duration in terms of both timing of
duration.

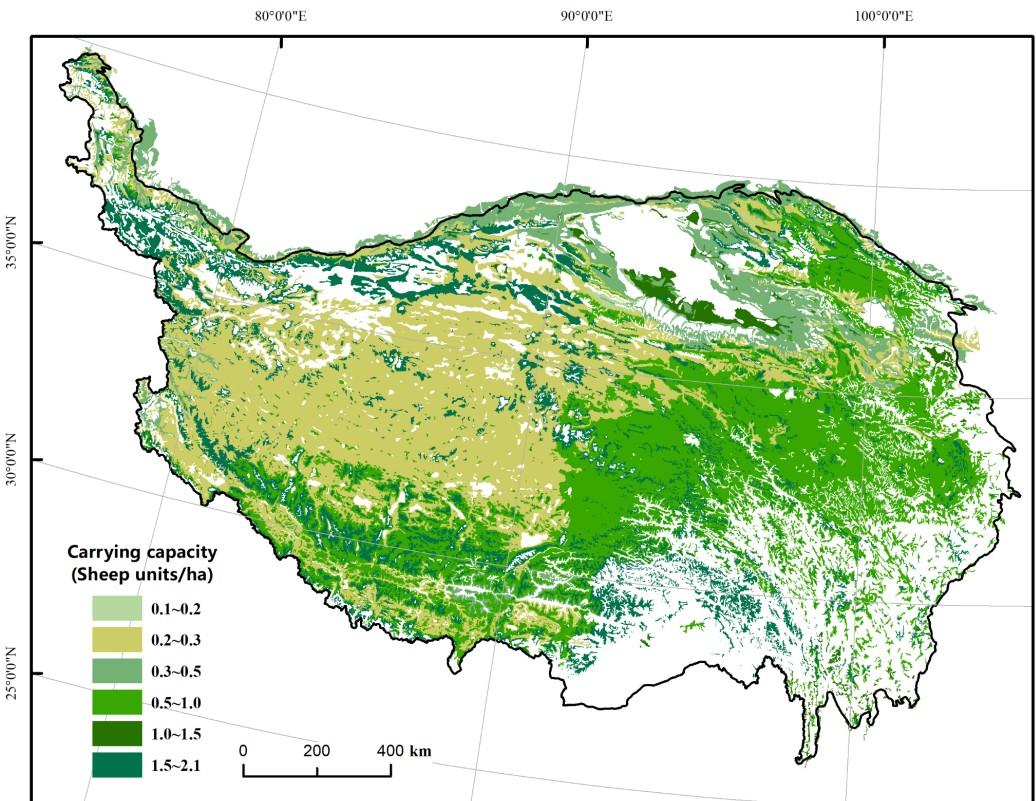

**Fig. A2 Spatial distribution of livestock exposure estimated from vegetation distribution**

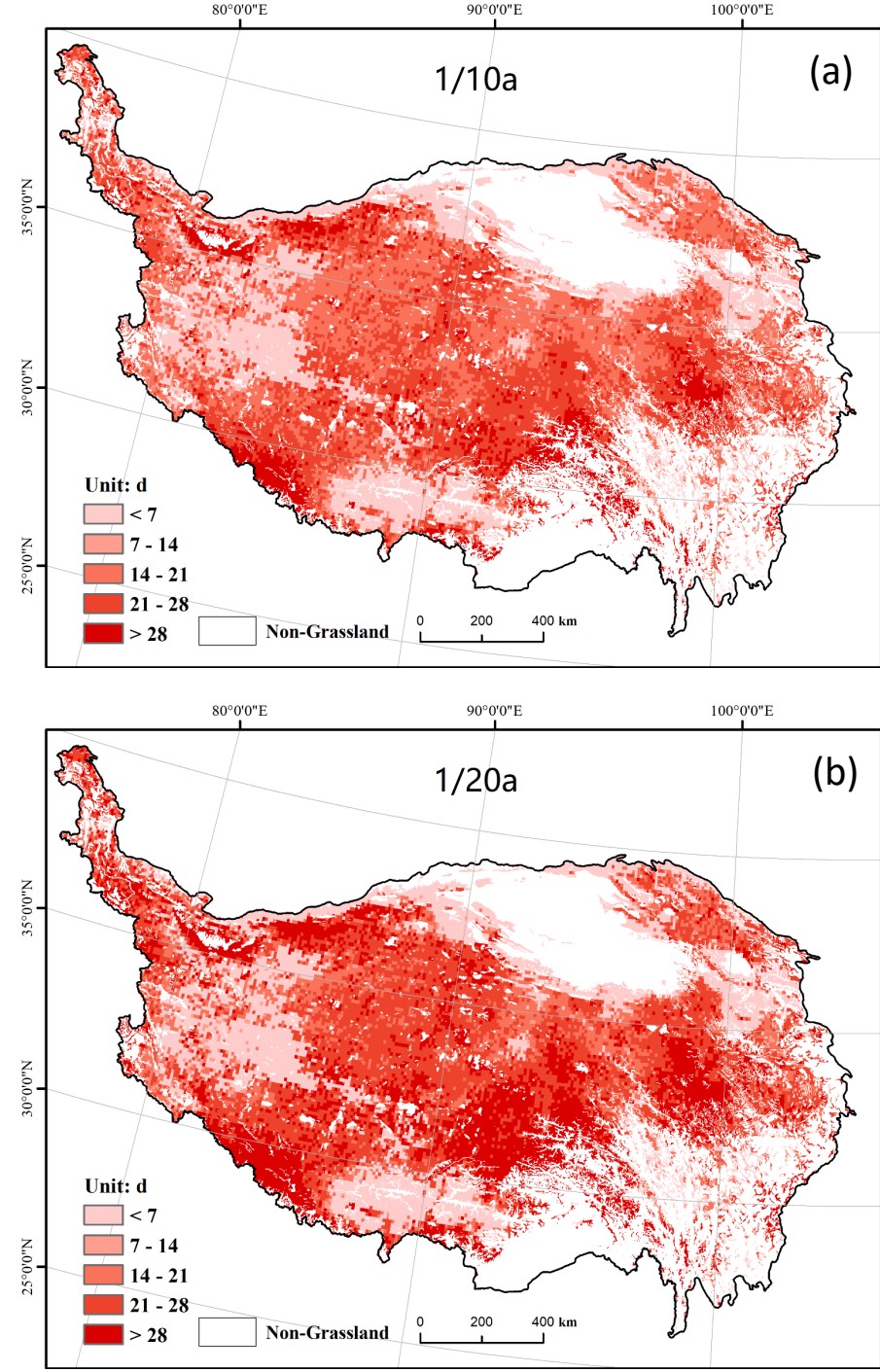

**Fig. A3 Gridded duration of a single disaster event by return period: (a) 1/10a; (b) 1/20a**

**Table A1 Look-up table of carrying capacity by grassland type in the QTP**

| Grassland type | Fresh grass yield (kg/ha) | Annual grazing rate (%) | Grassland required per sheep unit (ha/unit) | Carrying capacity (sheep unit/ha) |
|---|---|---|---|---|
| Alpine meadow | 1452 | 50 | 305.70 | 0.74 |
| Alpine steppe | 677 | 40 | 819.30 | 0.27 |
| Apline meadow-steppe | 689 | 45 | 745.20 | 0.30 |
| Alpine desert-steppe | 554 | 35 | 1077.30 | 0.21 |
| Apline desert | 519 | 30 | 988.95 | 0.23 |
| Temperate steppe | 3018 | 40 | 170.10 | 1.32 |
| Temperate desert | 683 | 30 | 1183.50 | 0.19 |
| Temperate desert-steppe | 611 | 35 | 840.75 | 0.27 |
| Lowland meadow | 3498 | 50 | 127.50 | 1.76 |
| Mountain meadow | 3879 | 55 | 132.30 | 1.70 |

Note: Figures were adapted from (Xin et al., 2011) and (Land Management Administration of Tibet Autonomous Region, 1994)

