# Peer review of "Event-based Probabilistic Risk Assessment of Livestock Snow Disasters in the Qinghai-Tibetan Plateau"

_Natural Hazards and Earth System Sciences, 2018_

## Referee Comment (RC1) · Anonymous Referee #1 · 22 Oct 2018

Reviewer' Comments

Manuscript Number: nhess-2018-182 Title: Probabilistic Risk Assessment of Livestock Snow Disasters in the Qinghai-Tibetan Plateau Journal: Natural Hazards and Earth System Sciences

General comments: Natural hazards that lead to disasters can cause tremendous impacts on societies, the environment, and economic wealth of the affected countries. Climate change will exacerbate existing challenges relating to livestock snow disaster risk. Adapting to climate change is a necessity for sensitive areas and those that are vulnerable to climate change such as the Qinghai-Tibetan Plateau. To investigate and better understand the risk of livestock snow disasters in the Qinghai-Tibetan Plateau is critical to towards sustainability of grassland animal husbandry and livelihood of the

herdsmen. This topic fits well with the mission and scope of Natural Hazards and Earth System Sciences. However, there are still many flaws in the current manuscript.

Special comments: 1. There are many abbreviated symbols in the paper such as QTP, PRA, SDD, BRT etc. Because of too many abbreviations, readers often get confused. It is suggested that a separate symbol page should be set up in front of the manuscript. 2. At present, there are many good quantitative methods for the study of snow disaster vulnerability and risk in alpine pastoral areas in particular on the Qinghai-Tibetan Plateau and Inner Mongolia Plateau. In the literature review in the "Introduction", the authors review this issue incompletely. The literature covered is also very limited. And main viewpoint may be biased. For example: Page 2, lines 19-20, "The first type employs an ordinal risk assessment framework in which the risk index is derived by integrating several indices representing different components of risk"; Page 2, lines 12-13, "The other risk assessment approach is quantitative, often called the probabilistic risk assessment (PRA), in which risk is measured with a probability distribution of socioeconomic losses (consequences)"; Page 2, lines 24-32, "However, studies applying PRA to livestock snow disasters have been limited. Bai et al. (2011) published one of the first trials in applying the PRA framework to a livestock snow disaster risk assessment. In their study, winter season (November to April of the preceding year) average daily snow depth was used to describe snow hazard intensity. Physical vulnerability, a function of livestock mortality rate in response to snow depth, was fitted using historical case data. Using annual average snow depth computed from satellite-retrieved data, return-period livestock mortality and mortality rates were derived as the final risk metrics. Based on their method, quantitative livestock snow disaster risks were mapped nationwide in China (Shi, 2011). The major flaw of this method was the mismatch between the event-based vulnerability function and annual measure of snow hazard. In another work focusing on Mongolia, a vulnerability function trained from a tree-based model was used, but still on an annual basis". 3. Page 3, lines 13-15, "Worldwide, the QTP suffers from some of the highest livestock snow disasters due to its large area of snow cover area, longlasting snow cover days, and nomadic grazing. This region is

also a hot spot in climate change. Quantitative risk assessments for the present day will likely be a significant source of information for disaster risk reduction". The above sentence should be moved to before line 5 on the second page. Delete lines 15-16 of the second page, "In addition, the framework can be adapted for livestock mortality in snow disasters in the context of future climate change analysis, and therefore support climate adaptation planning for local government and herding communities". 4. In the "Materials and Methods" section, the Qinghai-Tibet Plateau as case area, it is necessary to have a more comprehensive description of the geographical, environmental, social, and economic backgrounds of the QTP, especially the role of livestock in livelihood for local people. 5. We know the positive intervention of humans on the grassland ecosystem and that the grassland carrying capacity could be elevated with a reduction of harmful human activities (adverse effect), an increase of disaster prevention capacity. For example, the proportion of fenced pasture area to the total usable grassland (to show the capacity of grassland biomass to regenerate), the warm shed area per unit of livestock (to illustrate the capacity of livestock to prevent freezing disasters) and the proportion of sown grassland area to the total usable grassland (to descript the capacity of balancing forage supply and demand), accessibility of traffic and information (to depict the capacity of disaster response or prevention), if the above key factors are missing, in other words, if the authors do not emphasize the socio-system intervention for livestock snow disaster assessment, it will be very difficult to objectively assess the risk of snowstorms in livestock. 6. Page 4, line 10, "prevention capacity as measured by gross domestic production (GDP) of the underlying county", GDP as prevention capacity is not a scientific proxy, indeed, local fiscal revenue and the intensity of infrastructure construction in animal husbandry (including alpine grassland) are the key to reducing vulnerability and risk of livestock snow disaster. 7. Page 7, lines 3-6, the authors stated that "Historical snow disaster event data with the time of each event for each meteorological station were used to train the BRT model. These data were obtained from two sources. Records for 1980–2007 were obtained from W. Wang et al. (2013) while records from 2008–2015 were obtained from the China Meteorological Science Data Sharing Service System (CMSDS, http://data.cma.gov.cn)." However, are the identification criteria of the two snowstorm records sources consistent? 8. Page 8, in "2.3 Exposure", the herd size as a critical proxy of exposure, although the spatial distribution of livestock size can reflect the extent of snowstorm exposure of livestock, it is well known that the Qinghai-Tibetan Plateau has a vast area with obvious spatial differences, and the distribution density of livestock (the number of livestock per unit area) may be more scientifically and accurately describe the spatial feature of snowstorm exposure. 9. Similarly, page 8-9, in "2.4 Loss modelling", as one of loss index, GDP at county level is not consistent with the risk topic of livestock snow disaster. It is suggested that the added value of animal husbandry at county level should be adopted. 10. Page 9, lines 1-4, I don't understand that "County level GDP values were assigned to each grid within its boundary. We used constant GDP values for 2015 for two reasons. First, the results can be directly treated as a stationary time series for estimating the probability distribution, as the influence of prevention capacity improvement has been removed. Second, it meets the goal of risk assessment, to estimate the likelihood of potential loss in the near future". The GDP of each county changes with time. Dynamic GDP should be used instead of static GDP to predict the probability of loss, which is not consistent with reality. 11. The discussion part only deals with the content of spatial pattern (4.1Spatial patterns of livestock snow disaster risk in the QTP). As an important part of risk change, the characteristics of dynamic and temporal variation cannot be absent. Moreover, the authors do not pay attention to the causal relationship between risk and its influencing factors. 12. In "4.3 Risk-informed implications" Page 20, line 22, "Our results imply that the present level of preparedness in local regions are far from sufficient"; Page 21, line 5, "Due to the difficulty in improving prevention capacity, insurance schemes are needed to provide relief"; From the perspective of above mentioned sentences, this is not the inspiration of the article analysis, but the main existing problems. 13. The language of the manuscript is rather deficient and requires the re-editing of native speakers.

In summary, I suggest that the manuscript needs a major revision before considering

publication.

Please also note the supplement to this comment:
https://www.nat-hazards-earth-syst-sci-discuss.net/nhess-2018-182/nhess-2018-182-RC1-supplement.pdf
* * *

---

## Referee Comment (RC2) · Anonymous Referee #2 · 25 Oct 2018

**[General comments]**

"Probabilistic risk assessment of livestock snow disasters in the Qinghai-Tibetan Plateau" by Ye et al. applies boosted regression tree and general additive modeling methods to the snow disasters in the Qinghai-Tibetan Plateau, as an event-based evaluation, and the results are basically consistent with existing studies. The research topic is within the scope of the journal, but there are some substantial flaws in the study, which should be addressed. The major concerns are:

1: The advantage of event-based evaluation is not clear. Rather than hay preparation based on event-based analysis of this study, that based on annual-based analysis will work well when the intervals between the two events are very short (as time to prepare for the next event is not enough, preparation for annual basis is better, particularly when

**modeled annual frequency > 1).**

2. The authors say probabilistic analysis is one of the advantages of the study. However I consider a year-by-year evaluation is more sophisticated, and the probabilistic analysis used here is not necessarily an advantage, but a result of taking a simple evaluation dealing with what should be separately treated as one set of data. An effective PRA would be a result from a probabilistic function, not a result from treating various conditions as one case.

3: To evaluate livestock number by carrying capacity, and to evaluate carrying capacity by grassland type is questionable. The appropriateness of them should be more carefully discussed.

4. Explanation and discussion on Eq. 1 is not sufficient. The form of functions for each term in the right hand side, and the performance of the equation should be clearly presented.

**[Specific comments]**

Page 3 Line 1: Do you mean that the final metrics should always be mortality, not mortality rate? It looks the study considered mortality rate as the final metrics, and I think each study can use its own final metric.

Page 3 Line 13: Provide the rationale of "some of the highest livestock snow disasters". What is "highest", by the way? Largest damage? Highest frequency?

Page 3 Line 14: Provide literature for "This region is also a hot spot in climate change".

Page 4 Fig 1: It is difficult to understand the relationship between Timing, and Duration and Wind speed from the figure.

Page 4 Sect. 2.1: Fig 1 indicates GDP part is the vulnerability function, while Sect. 2.1 reads Eq. 1 (to derive mortality rate from GDP as well as hazard indicators). Which is true?

Page 4 Line 10 (eq 1): Present the detailed forms of s(Duration), s(Wind), s(P) and s(GDP). (all parameters of spline curves)

Page 4 Line 14: More detailed performance check is needed (not for InLR, but for LR). How large is RMSE? Is the error random or systematic? I want to see the scatterplot of observed and modelled values.

Page 5 Line 19: How the data of the previous days are used?

Page 6 Lines 12-13: How do you compromise when the two standards are different with each other?

Page 6 Line 14: Why since the last snow fall day?

Page 7 Lines 1-2: This sentence dose not explain why satellite is not used here.

Page 7 Lines 4-6: Are there no bias between the two data?

Page 7 Line 11: Explain the meanings of Ir and tc.

Page 7 Lines 13-14: How the number of the variables and prediction power is weighted? Any kind of criteria like AIC or BIC is used?

Page 7 Line 14: The cross validation is how many fold?

Page 7 Line 20: Is "prediction error" random? If they are systematic, to take average may not be a good solution.

Page 8 Lines 18-20: More careful discussion on the validity of this method is needed. I want to see the scatterplot of observed and estimated values.

Page 8 Lines 21-22: More careful discussion on the validity of this method is needed. I want to see the scatterplot of observed and estimated values.

Page 9 Line 9: How good is the performance of the equation?

Page 10 Line 4: Why SD, minWind and Pre were excluded?

СЗ

Page 10 Line 7: How relative contribution is calculated?

Page 10 Line 19 "well captured": p-values of 0.118 and 0.189 are not necessarily good (not statistically significant with 10% level). To check the model's representability more carefully, let us see not cumulative but probability density function before accumulation.

Page 12 Fig. 5: Better to present the annual total number of SDDs.

Page 13 Line 11 (also for Page 18 Line 1): It is obvious when the Gaussian approximation is used.

Page 13 Line 13: Fig 5b shows annual aggregate snow disaster duration? But the caption says "mean event duration".

Page 13 Lines 18-19: If "The distribution of annual average mortality rate is extremely positively skewed", the Gaussian kernel function (Page 9 lines 21-22) is not appropriate, is it? BTW, is it related to the dependent variable in Eq .1 is InLR, not LR?

Page 18 Table 1: What is the trend of actual herd size in QTP? To consider a static herd size is reasonable?

Page 18 Line 8: Why mortality becomes small by the constraint of herd size by carrying capacity?

Page 19 Lines 15-16: It is better to compare the modeled mortality with observed (historical) ones. Page 19 Lines 27-28: Is there no possibility that this study over-estimates?

Page 19 Sect. 4.2: As "more than one snow disaster a year is unlikely", annual evaluation is enough, isn't it? For me it looks to prepare hay based on annual evaluation is OK.

Page 21 Line 8 "two critical indices": Is this presented in the Result section?

Page 21 Lines 19-20: More careful evaluation is needed for the performance of herd

size estimation.

Page 22 Line 18: How the study can be applied for future? I consider that the method used here is not suitable when the climate is changing.

[Technical corrections]

Page 1 Lines 20,22: 1/20a -> 20 years (also for all similar expressions).

Page 5 Lines 4-6: Hard to understand. Too many "and"s. "its needs" -> "it needs"? Delete one of the two "provide"s?

Page 5 Line 5: Insert "in Eq. 1 " after Duration and Wind will be helpful.

Page 7 Line 4: "W. Wang et al" -> "Wang et al"

Page 9 Line 7: "although unlikely" should be rephrased with better expression.

Page 10 Line 14: Fig. 3 -> Fig. 4?

Page 13 Line 2: topology -> topography?

Pages 14-15 Fig 6: To be multi-colored like Fig 7 would be more reader-friendly.

Page 19 Line 22: There is no Table 2.

Page 19 Line 26 (also in Page 21 Line 28): higher -> longer.

Page 20 Line 8: Fig A2 -> A3?

---

## Author Comment (AC1) · 12 Dec 2018

**Reponses to Reviewer #1**

We greatly appreciate your comments and suggestions. We have make a careful plan to revise the manuscript according to what you have pointed out and believe that the quality of this manuscript will be improved as a result of the revision. We have included our detailed responses to each of your comments raised.

**General comments:**

Natural hazards that lead to disasters can cause tremendous impacts on societies, the environment, and economic wealth of the affected countries. Climate change will exacerbate existing challenges relating to livestock snow disaster risk. Adapting to climate change is a necessity for sensitive areas and those that are vulnerable to climate change such as the Qinghai-Tibetan Plateau. To investigate and better understand the risk of livestock snow disasters in the Qinghai-Tibetan Plateau is critical to towards sustainability of grassland animal husbandry and livelihood of the herdsmen. This topic fits well with the mission and scope of Natural Hazards and Earth System Sciences. However, there are still many flaws in the current manuscript.

**Special comments:**

1. There are many abbreviated symbols in the paper such as QTP, PRA, SDD, BRT etc. Because of too many abbreviations, readers often get confused. It is suggested that a separate symbol page should be set up in front of the manuscript.

RE: Thank you for your kind suggestion. As including a sperate symbol page in front of the manuscript is subjected to the decision of the editor, we will change all abbreviations to their full term throughout the manuscript to maximumly benefit the readers.

2. At present, there are many good quantitative methods for the study of snow disaster vulnerability and risk in alpine pastoral areas in particular on the Qinghai-Tibetan Plateau and Inner Mongolia Plateau. In the literature review in the "Introduction", the authors review this issue incompletely. The literature covered is also very limited. And main viewpoint may be biased. For example:

*Page 2, lines 19-20, "The first type employs an ordinal risk assessment framework in which the risk index is derived by integrating several indices representing different components of risk";*
*Page 2, lines 12-13, "The other risk assessment approach is quantitative, often called the probabilistic risk assessment (PRA), in which risk is measured with a probability distribution of socioeconomic losses (consequences)";*
*Page 2, lines 24-32, "However, studies applying PRA to livestock snow disasters have been limited. Bai et al. (2011) published one of the first trials in applying the PRA framework to a livestock snow disaster risk assessment. In their study, winter season (November to April of the preceding year) average daily snow depth was used to describe snow hazard intensity. Physical*

*vulnerability, a function of livestock mortality rate in response to snow depth, was fitted using historical case data. Using annual average snow depth computed from satellite-retrieved data, return-period livestock mortality and mortality rates were derived as the final risk metrics. Based on their method, quantitative livestock snow disaster risks were mapped nationwide in China (Shi, 2011). The major flaw of this method was the mismatch between the event-based vulnerability function and annual measure of snow hazard. In another work focusing on Mongolia, a vulnerability function trained from a tree-based model was used, but still on an annual basis".*

RE: Thank you for your kind suggestion. We did another round of careful literature search and review and found several important articles i.e. (Yeh *et al.*, 2014; Dong and Sherman, 2015; Miao *et al.*, 2016; Wei *et al.*, 2017). We have decided to add these references into the review section following your suggestion and suggestions from Reviewer #2 (comments 1 and 2).

References related to this comment:
Dong, S. and Sherman, R.: Enhancing the resilience of coupled human and natural systems of alpine rangelands on the Qinghai-Tibetan Plateau, Rangel. J., 37(1), i–iii, doi:10.1071/RJ14117, 2015.
Miao, L., Fraser, R., Sun, Z., Sneath, D., He, B. and Cui, X.: Climate impact on vegetation and animal husbandry on the Mongolian plateau: a comparative analysis, Nat. Hazards, 80(2), 727–739, doi:10.1007/s11069-015-1992-3, 2016.
Wei, Y., Wang, S., Fang, Y. and Nawaz, Z.: Integrated assessment on the vulnerability of animal husbandry to snow disasters under climate change in the Qinghai-Tibetan Plateau, Glob. Planet. Change, 157(March), 139–152, doi:10.1016/j.gloplacha.2017.08.017, 2017.
Yeh, E. T., Nyima, Y., Hopping, K. A. and Klein, J. A.: Tibetan Pastoralists' Vulnerability to Climate Change: A Political Ecology Analysis of Snowstorm Coping Capacity, Hum. Ecol., 42(1), 61–74, doi:10.1007/s10745-013-9625-5, 2014.

3. Page 3, lines 13-15, "*Worldwide, the QTP suffers from some of the highest livestock snow disasters due to its large area of snow cover area, longlasting snow cover days, and nomadic grazing. This region is also a hot spot in climate change. Quantitative risk assessments for the present day will likely be a significant source of information for disaster risk reduction*". The above sentence should be moved to before line 5 on the second page.
Delete lines 15-16 of the second page, *"In addition, the framework can be adapted for livestock mortality in snow disasters in the context of future climate change analysis, and therefore support climate adaptation planning for local government and herding communities".*

RE: These places will be revised according to your suggestion.

4. In the "Materials and Methods" section, the Qinghai-Tibet Plateau as case area, it is necessary to have a more comprehensive description of the geographical, environmental, social, and economic backgrounds of the QTP, especially the role of livestock in livelihood for local people.

RE: Per your suggestion, we have decided to add a sub-section exclusively introduce the QTP, including its geographical, environmental social and economic backgrounds, with emphasis on the role of livestock in livelihood for local people.

5. We know the positive intervention of humans on the grassland ecosystem and that the

grassland carrying capacity could be elevated with a reduction of harmful human activities (adverse effect), an increase of disaster prevention capacity. For example, the proportion of fenced pasture area to the total usable grassland (to show the capacity of grassland biomass to regenerate), the warm shed area per unit of livestock (to illustrate the capacity of livestock to prevent freezing disasters) and the proportion of sown grassland area to the total usable grassland (to descript the capacity of balancing forage supply and demand), accessibility of traffic and information (to depict the capacity of disaster response or prevention), if the above key factors are missing, in other words, if the authors do not emphasize the socio-system intervention for livestock snow disaster assessment, it will be very difficult to objectively assess the risk of snowstorms in livestock.

6. Page 4, line 10, "*prevention capacity as measured by gross domestic production (GDP) of the underlying county*", GDP as prevention capacity is not a scientific proxy, indeed, local fiscal revenue and the intensity of infrastructure construction in animal husbandry (including alpine grassland) are the key to reducing vulnerability and risk of livestock snow disaster.

7. Similarly, page 8-9, in "2.4 Loss modelling", as one of loss index, GDP at county level is not consistent with the risk topic of livestock snow disaster. It is suggested that the added value of animal husbandry at county level should be adopted.

RE: Above comments (5, 6, and 7) are related to each other and are responded together.

    We totally agree that it needs a thorough understanding of vulnerability to snow disaster before a good risk assessment carries out. You have offered important insights into local herders' coping capacity to snow disasters, and the factors that you mentioned (fenced pasture area, warm shed area, sown grassland, and accessibility) are critical in deciding vulnerability to snow disaster. These variables are valuable, but mostly only available for certain regions and can only be obtained from interview/survey. Per your suggestion, we checked again the statistical yearbooks, including the provincial statistical yearbooks of Qinghai and Tibet which date back to 1989, and the National County (City) Socioeconomic Statistical Yearbook (2000~ ), but found no indicators such as fenced pastures, warm shed areas, and sown grassland area.

    So we kept the strategy of using a proxy variable to indicate prevention capacity. Following your suggestion, we collected data on "fiscal revenue" (*Fiscal_Rev*) and "added value in animal husbandry" (*Value_Add*), which could be the first-best choices to denote prevention capacity. In addition, we also considered Fiscal Expenditure (*Fiscal_Exp*); and GDP per capita (*GDP_PC*). All the values were turned to 2015 Yuan. These variables were slightly-to-moderately correlated with GDP. The Pearson correlation coefficients between *Value_Add*, *Fiscal_Rev*, *Fiscal_Exp*, *GDP_PC* and *GDP* are: 0.336, 0.760, 0.420 and 0.223, respectively.

    We re-ran our generalized additive model (GAM) as shown in Eq (1) by replacing GDP with each of the four variables, and performed model diagnostics to check goodness-of-fit as well as response curves. Summary statistics of model runs are provided as below:

Table 1 Generalized additive model results by using different socioeconomic factors indicating prevention capacity

| $\ln LR$ $= s(Duration)$ $+ s(Wind) + s(P) +$ | R-sq.(adj) | Deviance explained | GCV | N(sample) | Significance level of the socioeconomic factor |
|---|---|---|---|---|---|

| | | | | | |
|---|---|---|---|---|---|
| $s(GDP)$ | 0.554 | 62.1% | 2.5105 | 79 | At 0.01 level |
| $s(Value\_Add)$ | 0.563 | 62.5% | 2.5508 | 73 | At 0.01 level |
| $s(Fiscal\_Rev)$ | 0.516 | 58.4% | 2.8392 | 73 | Not significant |
| $s(Fiscal\_Exp)$ | 0.524 | 58.4% | 2.7301 | 73 | At 0.01 level |
| $s(GDP\_PC)$ | 0.506 | 57.2% | 2.8561 | 74 | Not significant |

As shown in the table, variable *Fiscal_Rev* cannot improve the prediction of ln*LR* (natural logarithm of mortality rate). *Value_Add* is capable of deriving competing results. The response curves of the variables are also similar: ln*LR* showed downward slope with each of the three variables, indicating decreasing loss rate in response to enhanced prevention capacity.

Given above analysis, together with your suggestion, we have decided to use value added of animal husbandry in our vulnerability function, and update all our results throughout the manuscript.

8. Page 7, lines 3-6, the authors stated that "*Historical snow disaster event data with the time of each event for each meteorological station were used to train the BRT model. These data were obtained from two sources. Records for 1980–2007 were obtained from W. Wang et al. (2013) while records from 2008–2015 were obtained from the China Meteorological Science Data Sharing Service System (CMSDS, http://data.cma.gov.cn)*." However, are the identification criteria of the two snowstorm records sources consistent?

RE: This comment is high related to Reviewer #2's comment (Page 7 Lines 4-6: Are there no bias between the two data?). According to (Wang *et al.*, 2013), the data for period of 1980-2007 were obtained from the yearbooks of meteorological disasters. Therefore, these data were originally recorded by provincial meteorological administrations officially, and published as a collection in yearbooks. The data for 2008-2015 were directly obtained from China Meteorological Administration in digital format. Therefore, they are both from official records from meteorological administration, the standards in identifying snow disasters are the same according to local Meteorological Administration officials.

In the revision, we have decided to add the information to clarify the potenail bias between two data is very limited. "*Records for 1980–2007 were a collection of snow disaster records published in 6 provincial meteorological yearbooks neighboring the Plateau (Wang et al., 2013b). Records from 2008–2015 were obtained from the China Meteorological Science Data Sharing Service System (CMSDS, http://data.cma.gov.cn). Records in both datasets are official observations by the meteorological administrations and are consistent with each other in terms of observation standards*."

9. Page 8, in "2.3 Exposure", the herd size as a critical proxy of exposure, although the spatial distribution of livestock size can reflect the extent of snowstorm exposure of livestock, it is well known that the Qinghai-Tibetan Plateau has a vast area with obvious spatial differences, and the distribution density of livestock (the number of livestock per unit area) may be more scientifically and accurately describe the spatial feature of snowstorm exposure.

RE: Thank you for your suggestion. We have changed our term from "herd size" to "herd density" where exposure is discussed. In the exposure map we derived (Fig. A2), the unit has already been (Sheep unit/ha). Accordingly, we will update the risk metrics map in terms of mortality (Fig.7) to show the loss measured with sheep units/km$^2$.

10. Page 9, lines 1-4, I don't understand that *"County level GDP values were assigned to each grid within its boundary. We used constant GDP values for 2015 for two reasons. First, the results can be directly treated as a stationary time series for estimating the probability distribution, as the influence of prevention capacity improvement has been removed. Second, it meets the goal of risk assessment, to estimate the likelihood of potential loss in the near future"*. The GDP of each county changes with time. Dynamic GDP should be used instead of static GDP to predict the probability of loss, which is not consistent with reality.

RE: Thanks for your comment. In our vulnerability function, GDP (has been changed to value added in animal husbandry per your comment) was used as an indicator of prevention capacity. Therefore, whether to use historical dynamic value or a static present value essentially depends on our purpose of analysis.

  1) If we are modeling actual historical losses for model calibration and verification purposes, historical dynamic value should be used (for such discussion, please refer to the response to reviewer#2's comment regarding Page 19 Lines 15-16 and Page 19 Lines 27-28).

  2) If we are assessing livestock risk (the probability of potential loss) for the next couple of years, then using present-day prevention capacity would be a better choice than using historical prevention capacity. Then, our risk assessment effort tries to answer "if historical events occur in nowadays prevention capacity, how would the probability distribution of the loss will be". Correspondingly, the risk metrics would be meaningful for prevention planning and insurance implications because we are considering the near future given today's situation.

  Technically, to fit a probability distribution from samples of loss requires that the sample data must be at least stationary in its mean and variance, so as to remove any technical, environment, or prevention capacity change effect. This is the reason in many risk assessment research, historical loss must be "detrended" before it was fit (Maddala, 1977; Lobell and Burke, 2010; Ye *et al.*, 2015). In our case, as GDP (or value added in animal husbandry) keeps growing along the time, modeled losses based on historical dynamic GDP (or value added in animal husbandry will contain obvious trend and therefore cannot be used directly to fit any probability distribution.

  In order to better explain the difference, we modeled annual winter losses (from September ~ June of the next year) using both historical (dynamic) and static (2015) value added of animal husbandry, and the time series are shown below (Figure 1).

[Figure]

Figure 1 Modeled total livestock loss over the Tibetan Plateau. The blue time series was losses modeled using constant *Value_Add* of year 2015, assuming present-day prevention capacity (static). The orange time series was losses modeled using *Value_Add* of historical values (dynamic), assuming historical prevention capacity. All the losses are for a specific snow disaster season from September to the next June rather than a civil year.

Loss modeled using historical value added of animal husbandry (*Value_Add*, the orange time series in Figure 1) obvious contained trends. It showed an obvious downward trend. Fitting a probability distribution would over-estimate the size of risk. Using a static present value of one recent year, it technically meets the requirement of a stationary time series (the blue trend line is flat in Figure 1). Then losses derived from our event-based model are derived with the prevention level of historical period. Their difference demonstrates the effect of improved prevention capacity (which might echoes the Comment #1, and provides meaningful discussion for your Comment #11).

In the revision, more explanation about the difference of the modeled losses using static and dynamic socioeconomic data will be provided.

11. The discussion part only deals with the content of spatial pattern (4.1Spatial patterns of livestock snow disaster risk in the QTP). As an important part of risk change, the characteristics of dynamic and temporal variation cannot be absent. Moreover, the authors do not pay attention to the causal relationship between risk and its influencing factors.

RE: Thank you for your comment. We have prepared to add a subsection in the result section show temporal changes of livestock mortality derived from our model, and enrich our discussion by comparing it to historical losses observed.

The discussion over the causal relationship between risk and its influencing factors can be partly be done by comparing the dynamics of the modeled loss with historical prevention capacity and static present-day prevention capacity, as shown in Figure 1. We will also added discussion about the role of improved prevention capacity in section 4.3 Risk-informed implications. ─

12. In "4.3 Risk-informed implications" *Page 20, line 22, "Our results imply that the present level of preparedness in local regions are far from sufficient";*
*Page 21, line 5, "Due to the difficulty in improving prevention capacity, insurance schemes are needed to provide relief";*
From the perspective of above mentioned sentences, this is not the inspiration of the article

analysis, but the main existing problems.

RE: Thank you for your comment. We plan to revise the discussion section by 1) better emphasizing the advantage of event-based probabilistic risk assessment, particularly its capability of providing quantitative measure of preparedness capacity using return-period values, and 2) discussing the contribution of enhanced prevention capacity as suggested by your comment #11.

13. The language of the manuscript is rather deficient and requires the re-editing of native speakers.

RE: We will send the paper for professional proof-reading service before re-submission.

**References**

Dong, S. and Sherman, R. (2015) 'Enhancing the resilience of coupled human and natural systems of alpine rangelands on the Qinghai-Tibetan Plateau', *Rangeland Journal*, pp. i–iii. doi: 10.1071/RJ14117.

Lobell, D. B. and Burke, M. B. (2010) 'On the use of statistical models to predict crop yield responses to climate change', *Agricultural and Forest Meteorology*. Elsevier B.V., 150(11), pp. 1443–1452. doi: 10.1016/j.agrformet.2010.07.008.

Maddala, G. S. (1977) 'Introduction to Econometricsc', *Macmillan Publishing CompanyNew York*. doi: 10.1098/rspa.1963.0204.

Miao, L., Fraser, R., Sun, Z., Sneath, D., He, B. and Cui, X. (2016) 'Climate impact on vegetation and animal husbandry on the Mongolian plateau: a comparative analysis', *Natural Hazards*. Springer Netherlands, 80(2), pp. 727–739. doi: 10.1007/s11069-015-1992-3.

Wang, W., Liang, T., Huang, X., Feng, Q., Xie, H., Liu, X., Chen, M. and Wang, X. (2013) 'Early warning of snow-caused disasters in pastoral areas on the Tibetan Plateau', *Natural Hazards and Earth System Science*, 13(6), pp. 1411–1425. doi: 10.5194/nhess-13-1411-2013.

Wei, Y., Wang, S., Fang, Y. and Nawaz, Z. (2017) 'Integrated assessment on the vulnerability of animal husbandry to snow disasters under climate change in the Qinghai-Tibetan Plateau', *Global and Planetary Change*, 157(March), pp. 139–152. doi: 10.1016/j.gloplacha.2017.08.017.

Ye, T., Nie, J. L., Wang, J., Shi, P. J. and Wang, Z. (2015) 'Performance of Detrending models for crop yield risk assessment: evaluation with real and hypothetical yield data', *Stochastic Environmental Research and Risk Assessment*, 29(1), pp. 109–117. doi: 10.1007/s00477-014-0871-x.

Yeh, E. T., Nyima, Y., Hopping, K. A. and Klein, J. A. (2014) 'Tibetan Pastoralists' Vulnerability to Climate Change: A Political Ecology Analysis of Snowstorm Coping Capacity', *Human Ecology*, 42(1), pp. 61–74. doi: 10.1007/s10745-013-9625-5.

---

## Author Comment (AC2) · 12 Dec 2018

**Reponses to Reviewer #2**

We greatly appreciate your comments and suggestions. We have make a careful plan to revise the manuscript according to what you have pointed out and believe that the quality of this manuscript will be improved as a result of the revision. We have included our detailed responses to each of your comments raised.

**[General comments]**

"Probabilistic risk assessment of livestock snow disasters in the Qinghai-Tibetan Plateau" by Ye et al. applies boosted regression tree and general additive modeling methods to the snow disasters in the Qinghai-Tibetan Plateau, as an event-based evaluation, and the results are basically consistent with existing studies. The research topic is within the scope of the journal, but there are some substantial flaws in the study, which should be addressed. The major concerns are:

1: The advantage of event-based evaluation is not clear. Rather than hay preparation based on event-based analysis of this study, that based on annual-based analysis will work well when the intervals between the two events are very short (as time to prepare for the next event is not enough, preparation for annual basis is better, particularly when modeled annual frequency > 1).

Page 19 Sect. 4.2: As "more than one snow disaster a year is unlikely", annual evaluation is enough, isn't it? For me it looks to prepare hay based on annual evaluation is OK.

2. The authors say probabilistic analysis is one of the advantages of the study. However I consider a year-by-year evaluation is more sophisticated, and the probabilistic analysis used here is not necessarily an advantage, but a result of taking a simple evaluation dealing with what should be separately treated as one set of data. An effective PRA would be a result from a probabilistic function, not a result from treating various conditions as one case.

RE: Above questions are inter-related and are therefore responded together.

We totally agree that annual based modeling has its own advantages, but the event-based approach is also very important. By nature, livestock snow disaster can occur multiple times in a winter, and the losses would accumulate event by event. Capturing the details of event occurrence and intensity are important for following aspects:

1) ln$LR$ (natural logarithm of loss rate) shows a concave relationship with disaster duration (can be found later in details in response to your comment #4). Therefore, the total loss of one event lasting 30 days and two-event in one winter lasting 15 days each will be totally different. Knowing only the aggregate duration in a year cannot reflect such important details.

2) Although presently our data shows that historically it has been less likely to have more than 1 snow disaster in a winter for most parts of the Qinghai-Tibet. But if we are considering a method that can be generalized, then we need to consider the possibility of changing frequency and intensity, and the capacity to model it, particularly with the observed and

projected slight increase in precipitation on the Plateau (Kuang and Jiao, 2016; GUO, SUN and YU, 2018). Using the annual loss approach cannot capture such changes.

3) From the perspective of risk-informed action, annual evaluation (i.e. potential aggregate duration) would be temporarily sufficient for preparedness by the government and community. But it can hardly work for risk-transfer mechanisms as insurance schemes were mostly based on events. This is also the critical reason that catastrophe risk models are mostly built on event-basis (Michel-Kerjan *et al.*, 2013).

For the advantage of this study, we totally agree with you that a year-by-year evaluation is more sophisticated (than a simple statistical analysis). Our probabilistic analysis was built on event-by-event loss modeling, which is even more sophisticated than year-by-year evaluation. Benefited from your comment, we plan to add these discussions into the introduction section and discussion section to highlight the advantage of event-based modeling instead of probabilistic analysis.

3: To evaluate livestock number by carrying capacity, and to evaluate carrying capacity by grassland type is questionable. The appropriateness of them should be more carefully discussed.
Page 8 Lines 18-20: More careful discussion on the validity of this method is needed. I want to see the scatterplot of observed and estimated values.
Page 8 Lines 21-22: More careful discussion on the validity of this method is needed. I want to see the scatterplot of observed and estimated values.
Page 21 Lines 19-20: More careful evaluation is needed for the performance of herd size estimation.

RE: Thank you very much for your comment. Above comments are related and therefore responded together.

1) The appropriateness of evaluating livestock number by carrying capacity

We first need to clarify that we are trying to use carrying capacity to evaluate livestock number EXPOSED to snow disaster, but not the total livestock number. In the study area, only livestock grazing on grassland in pastoral and agro-pastoral regions (central to western part of Tibet Plateau) are exposed to snow disaster. Livestock kept in livestock farms in agricultural regions (mostly the eastern and low altitude parts of the Plateau) are not. When estimating exposure, we were trying to estimate the livestock number grazing on grassland.

The livestock number grazing on grassland is essentially determined by livestock carrying capacity, which exactly defines the maximum livestock number that can graze in a specific area of grassland, given local forage productivity and grazing style. Since 2011, the Tibet Autonomous region started the program of " subsidy and award policies for grassland ecological conservation efforts" to reduce the number of livestock and conserve grassland. In 2014-15, news articles intensively reported that "Tibet has basically reached forage-livestock balance" (government release, http://www.gov.cn/xinwen/2014-05/12/content_2677946.htm, in Chinese). In other words, livestock number grazing on grassland is highly consistent to the carrying capacity officially designated.

2) The appropriateness of evaluating carrying capacity by grassland type

Evaluating carrying capacity by grassland type is the way that adopted by both official calculation technical manual (*Ministry Standard of Calculation of Rangeland Carrying Capacity issued by Ministry of Agriculture of China*, NY/T 635-2015) and academic studies regarding forage-livestock balance, i.e. (Xin *et al.*, 2011). Grassland type essentially determines the key values of evaluating carrying capacity, i.e. forage regrowth percentage, proper utilization rate of rangeland (of different grazing seasons), conversion coefficient of standard hay. Using grassland type as a identifier of different carrying capacity is also the widely adopted way of presenting the evaluation results.

3) Verification of livestock number estimated

We obtained official release of carrying capacity by Tibet Autonomous Region from *Statistical Materials of Grassland Resource and Ecology of Tibet Autonomous Region* published by the Department of Agricultural and Pastoral of Tibet Autonomous Region in year 2011. We used zonal statistics to derive county-level carrying capacity estimated by our method, and compared with the official release (Figure 1).

[Figure]

Figure 1 Estimated and officially released livestock carrying capacity in Tibet Autonomous Region

As the above figure shows, the estimated and officially released livestock carrying capacity basically agree with each other.

In the revision, we plan to 1) clarify that we are estimating the herd size exposed to snow disaster (livestock number on free grazing) rather than total herd size; and 2) further explain the appropriateness of estimating livestock number of carrying capacity, and estimating carrying capacity by grassland type.

4. Explanation and discussion on Eq. 1 is not sufficient. The form of functions for each term in the right hand side, and the performance of the equation should be clearly presented.
Page 4 Line 10 (eq 1): Present the detailed forms of s(Duration), s(Wind), s(P) and s(GDP). (all parameters of spline curves)

Page 4 Line 14: More detailed performance check is needed (not for lnLR, but for LR). How large is RMSE? Is the error random or systematic? I want to see the scatterplot of observed and modelled values.

RE: Thank you for these questions and comments. In light of the suggestion from Reviewer #1 (please refer to Reviewer#1 Comments 5,6, and 9), we have updated the model by using value added of animal husbandry of the underlying county instead of GDP as the indicator of prevention capacity. We have updated Eq(1) accordingly to: $\ln LR = s(Duration) + s(Wind) + s(P) + s(Value\_Add)$. Details of the updated model are provided below.

(1) Model-fitting statistic:

```
Family: gaussian
Link function: identity

Formula:
lnLR ~ s(Duration) + s(Value_Add, k = 4) + s(maxWind) + s(P)

Parametric coefficients:
            Estimate Std. Error t value Pr(>|t|)
(Intercept)   0.5300     0.1708   3.103  0.00287 **
* * *
Signif. codes:  0 '***' 0.001 '**' 0.01 '*' 0.05 '.' 0.1 ' ' 1

Approximate significance of smooth terms:
               edf Ref.df     F  p-value
s(Duration)  5.626  6.705 9.706 1.96e-08 ***
s(Value_Add) 1.000  1.000 8.878  0.00407 **
s(maxWind)   2.732  3.440 2.747  0.04463 *
s(P)         1.000  1.000 2.809  0.09864 .
* * *
Signif. codes:  0 '***' 0.001 '**' 0.01 '*' 0.05 '.' 0.1 ' ' 1

R-sq.(adj) =  0.563   Deviance explained = 62.5%
GCV = 2.5508  Scale est. = 2.1593     n = 74
```

Figure 2 Fitting statistic of the updated GAM model

(2) The response curves (spline curves)

[Figure]

Figure 3 Response curves of the updated GAM model

Its response curves (Figure 3) indicate that: (1) ln*LR* is increasing with snow disaster duration. Duration up to 15-18 d is a critical period that mortality will increase rapidly. (2) ln*LR* decreases with value added of animal husbandry (*Value_Add*), indicating the effect of stronger

prevention capacity in reducing mortality, i.e. government expenditure in reserving hay for preparedness, and subsidy to herders to build/enlarge warm sheds. (3) An inverted-U shaped relationship between daily maximum wind speed and ln*LR*. The up-slope part indicates the increasing stress of stronger wind on livestock, but the down-slope part (beyond 5-6 m/s) indicate herder's reaction to stop free-grazing and keep herds in shelters (Wu *et al.*, 2007). (4) ln*LR* decreases with growing season precipitation. Larger growing season precipitation indicates more abundant food for livestock in summer, and therefore better body-condition in resisting low temperature and lack of food in snow disaster times.

(3) Model performance diagnostics

We performed 10-fold cross validation, and found RMSE, MAE and ME for the model were 1.747, 1.325, and -0.002, respectively.

The performance diagnostics charts of the model (Figure 4) indicate that 1) QQ plot is very close to a straight line, suggesting our distributional assumption of normality about ln*LR* is reasonable. 2) The variance is approximately constant as the mean increases. 3) The histogram of residuals appears approximately consistent with normality. 4) the response against fitted values show a positive linear relationship in the scatter plot.

[Figure]

Figure 4 Performance diagnostics charts for the updated GAM model

In the revision, we shall supply more details about the model fitted in to the manuscript. In addition, model details, including the fitting statistics, response curves, and performance diagnostics charts (Figure 2~Figure 4) will be provided in supplementary document for reference.

[Specific comments]

Page 3 Line 1: Do you mean that the final metrics should always be mortality, not mortality rate? It looks the study considered mortality rate as the final metrics, and I think each study can use its own final metric.

RE: Thank you for your comment. The sentence was misleading. In the revision, we plan to delete the latter part of this sentence and kept on "*Compared to earlier works, they successfully extended the framework to future climate change analysis*".

Page 3 Line 13: Provide the rationale of "some of the highest livestock snow disasters". What is "highest", by the way? Largest damage? Highest frequency?

RE: We intended to say "one of the regions with the highest risk" in terms of both high frequency and large damage. In the revision, we plan to this sentence to "…*the QTP is one of the regions that has suffered the most from livestock snow disasters due to its large area of snow cover area, long-lasting snow cover days, and nomadic grazing.* "

Page 3 Line 14: Provide literature for "This region is also a hot spot in climate change".

RE: References will be provided into the text as suggested in the revision, i.e. (Diffenbaugh and Giorgi, 2012; Gu *et al.*, 2014).

Diffenbaugh, N. S. and Giorgi, F.: Climate change hotspots in the CMIP5 global climate model ensemble., Clim. Change, 114(3–4), 813–822, doi:10.1007/s10584-012-0570-x, 2012.

Gu, H., Yu, Z., Wang, J., Ju, Q., Yang, C. and Fan, C.: Climate change hotspots identification in China through the CMIP5 global climate model ensemble, Adv. Meteorol., 2014, doi:10.1155/2014/963196, 2014.

Page 4 Fig 1: It is difficult to understand the relationship between Timing, and Duration and Wind speed from the figure.

RE: The figure presented in the online document was not complete due to unknown technical reasons when generating .pdf files. The correct figure should be:

[Figure]

Figure 5 Corrected technical flow of the study (Fig 1 in the manuscript)

In the modeling process, we derive timing (both starting and ending dates) from the "snow disaster event identifier/ simulator". Then the duration of the event (days between the starting and ending dates) and the wind speed during this period can be derived (as we have daily wind speed data). In the revision, we will correct the figure and add necessary explanation about the flow.

Page 4 Sect. 2.1: Fig 1 indicates GDP part is the vulnerability function, while Sect. 2.1 reads Eq. 1 (to derive mortality rate from GDP as well as hazard indicators). Which is true?

RE: GDP was one out of the four inputs of the vulnerability function (others were hazard indicators), which was used to indicate prevention capacity. In the revision, we have updated Fig. 1 in the manuscript by moving the vulnerability function into box "Event loss rate modeling" to keep it consistent with Eq. 1.

Page 5 Line 19: How the data of the previous days are used?
Page 6 Line 14: Why since the last snow fall day?
RE: The two comments are related and are therefore responded together.

We used average daily precipitation (snowfall) since the last snowfall day (the last day with effective precipitation) to represent the diminishing impact/pressure from snowfall as time elapses. In general, the larger amount of snowfall, the longer the impact of the snowfall would be given identical temperature conditions – it would take longer time to thaw. Similarly, the closer a day to the snowfall day, the more likely it would be identified as a "snow disaster" day. It worked especially for deciding whether two snow fall days should be regarded as parts of one single event or two independent events.

Page 6 Lines 12-13: How do you compromise when the two standards are different with each other?
RE: Thank you for your comment. The two standards, although designed for different regions, are not conflicting with each other on the key indicators (Table 1), but shares similar indicators (i.e. continuous days of perpetual snow cover, or number of consecutive snow fall days since the last snow fall day).

Table 1 Variables considered in national and industrial standards for snow disaster

| Snow Disaster Grades in Grazing Regions of China (GB/T20482-2017) | Meteorological Grades of Urban Snow Hazards (QX/T 178-2013) |
|---|---|
| Snow depth (cm) | Cumulative snowfall (mm) |
| Grass height (cm) | Maximum daily snowfall (mm) |
| Continuous days of snow cover (d) | Snow depth (cm) |
| % of grassland covered by snow (%) | Number of consecutive snowfall days (since the last snowfall day) |
|  | Daily lowest temperature (degree C) |
|  | Windspeed (m/s) |
|  | Minimum relative humidity *(%) |

Page 7 Lines 1-2: This sentence dose not explain why satellite is not used here.

RE: Thank you for your comment. Data need from satellite imagery is mainly the daily snow cover ("yes/no" data) and snow cover rate (% area covered by snow). One of the major purpose of this study is to develop a probabilistic risk assessment framework for risk assessment future climate change scenarios. For such a purpose, our input variables used in the framework must be available in climate projections. As far as we know, there are no projections of daily snow cover for the future. This is one of the main reasons that we did not consider it.

In the revision, we will try to make this point clear to the readers.

Page 7 Lines 4-6: Are there no bias between the two data?

RE: A similar question has been raised by Reviewer #1 (comment #7). According to (Wang *et al.*, 2013), the data for period of 1980-2007 were obtained from the yearbooks of meteorological disasters. Therefore, these data were originally recorded officially by provincial meteorological administrations, and published as a collection in books. The data for 2008-2015 were directly obtained form China Meteorological Administration in digital format. Therefore they are both from official records from meteorological administration, and the standards in identifying snow disasters are the same. However, we cannot perform a bias check as the data from two different sources do not share any overlapping period.

In the revision, we will supply information to clarify that the potenail bias between two data is very limited.

Page 7 Line 11: Explain the meanings of lr and tc.

RE: Boosted Regression Trees have two important parameters that need to be specified by the user. Tree complexity (*tc*) controls the number of splits in each tree. A *tc* value of 1 results in trees with only 1 split, and means that the model does not take into account interactions between environmental variables. A *tc* value of 2 results in two splits and so on. Learning rate (*lr*) determines the contribution of each tree to the growing model. As small value of *lr* results in many trees to be built. These two parameters together determine the number of trees that is required for optimal prediction. The aim is to find the combination of parameters that results in the minimum error for predictions.

In the revision, we will supply these information into the text.

Page 7 Lines 13-14: How the number of the variables and prediction power is weighted? Any kind of criteria like AIC or BIC is used?
Page 10 Line 4: Why SD, minWind and Pre were excluded?
RE: SD, minWInd and Pre were excluded due to their modest contribution in explaining the response variable. BRT uses a process of variable selection analogous to backward selection in regression. It drops the least important (in terms of relative influence) predictor, then re-fits the model and sequentially repeating the process (Elith, Leathwick and Hastie, 2008). In each step, after the removal of one predictor, the change in predictive deviance is computed relative to that obtained when using all predictors. Finally, a list containing the mean change in deviance and its standard error as a function of the number of variables removed will be returned

(Hijmans *et al.*, 2011). From the list, the optimal number of variables to drop can be identified, i.e. the number of variables that yield the minimum predictive deviance.

In the revision, we will supply more information about the mechanism of variable selection and model simplification.

Page 7 Line 14: The cross validation is how many fold?

RE: We used a 10-fold cross-validation. This information will be supplied in the revision.

Page 7 Line 20: Is "prediction error" random? If they are systematic, to take average may not be a good solution.

RE: The CV estimates of prediction error (predictive deviance in BRT models, Elith et al. 2008) of our BRT model indicates that the error is random.

Page 9 Line 9: How good is the performance of the equation?

RE: This equation is derived using logic reasoning: if a share of livestock died in one event, then the actual herd size exposed to the next event should be reduced correspondingly. As we rarely have two or more subsequent events in one year and one place in our historical records, we cannot evaluate the performance of this equation. But it is logically correct.

Page 10 Line 7: How relative contribution is calculated?

RE: In BRT, the relative importance are calculated based on the number of times a variable is selected for splitting, weighted by the squared improvement to the model as a result of each split, and averaged over all trees. The relative importance for each variable is scaled so that the sum adds to 100 (Elith, Leathwick and Hastie, 2008).

Page 10 Line 19 "well captured": p-values of 0.118 and 0.189 are not necessarily good (not statistically significant with 10% level). To check the model's representability more carefully, let us see not cumulative but probability density function before accumulation.

RE: We agree with your that we may not use "well captured". But we want to double confirm about the meaning of "not statistically significant with 10% level". The null hypothesis of the Two-sample Kolmogorov–Smirnov test is: "The two samples come from a common distribution", and the alternative hypothesis is: "The two samples do not come from a common distribution"

(https://www.itl.nist.gov/div898/software/dataplot/refman1/auxillar/ks2samp.htm). Our test statistics indicated "not statistically significant at 10% level", and therefore we failed to reject the alternative hypothesis and had to believe that the two samples (observed and predicted) were from a common distribution. It said that our prediction had captured the statistical feature of the observed duration (both event and annual). We posted the probability densities of these variables below, and actually they show similar thing with the cumulative density function:

[Figure]

Figure 6 probability densities of observed and predicted snow disaster durations

In the revision, we will change "well captured" to "captured" following your suggestion.

Page 12 Fig. 5: Better to present the annual total number of SDDs.

Page 13 Line 13: Fig 5b shows annual aggregate snow disaster duration? But the caption says "mean event duration".

RE: Two questions above are related and responded together.

We believe that the reviewer has been discussing Fig. 5 (b). In the revision, we will follow your suggestion and present the annual aggregate duration ("annual total number of SDDs"). Correspondingly, the description on Page 13 will be updated to keep consistency with the figure.

Page 13 Line 11 (also for Page 18 Line 1): It is obvious when the Gaussian approximation is used.

Page 13 Lines 18-19: If "The distribution of annual average mortality rate is extremely positively skewed", the Gaussian kernel function (Page 9 lines 21-22) is not appropriate, is it? BTW, is it related to the dependent variable in Eq .1 is lnLR, not LR?

RE: Thank you for your question. The two comments are related and are responded together. There are three points to clarify:

1) We did not use Gaussian approximation over the simulated annual loss rates, which is a parametric and symmetric distribution function. In stead, we used a non-parametric approach called kernel density function (Page 9 lines 21-24). The kernel density approach does not specify any functional form and so it is flexible to capture probability densities of different degrees of skewness.

2) Page 13 Lines 18-19 was describing the distribution of annual average mortality rate of different grids (spatial locations), but not the distribution of annual mortality rate for any specific location. "Extremely positively skewed" means that the grids/regions of high mortality risk take only a small portion of all places on the Plateau, but their annual average mortality rates were much higher than those of other grids. It is not related to the distributional assumption of dependent variable LR.

3) LR is also positively skewed. Only its natural logarithm (lnLR) exhibits normality.

Page 18 Table 1: What is the trend of actual herd size in QTP? To consider a static herd size is reasonable?

RE: The trend of herd size as a total of Qinghai and Tibet is provided below in Figure 7. As shown in the figure, number of cattle has not changed much for both regions. Number of sheep in Qinghai has not changed much since 2006, but that of Tibet has been decreasing for recent years, mainly due to the forage-livestock balance policy. In terms of aggregate size (1 cattle = 5 sheep units), Qinghai remains quite stable since 2006, and Tibet keeps dropping rapidly up to year 2014. Once it drops to the upper limit of carrying-capacity, it is very likely to keep nearly constant (please also refer to the discussion on your comment #3).

[Figure]

Figure 7 Change of herd size summed from Qinghai and Tibet

In our results, we will add discussion to this point.

Page 18 Line 8: Why mortality becomes small by the constraint of herd size by carrying capacity?

RE: This sentence was misleading. We intended to say that the mortality (sheep units) was small (mostly below 10 per grid), and the main reason was that the estimated carrying capacity was small. In the revision, we will change this sentence to: "*Mortality appears small in Fig. 7, generally several sheep units/km². However, when aggregated at the prefecture level, mortality remained considerable (Table 1).*"

Page 19 Lines 15-16: It is better to compare the modeled mortality with observed (historical) ones.

Page 19 Lines 27-28: Is there no possibility that this study overestimates?

RE: The two questions above are related to each other and are responded together.

To further test the performance of the model, we have re-run the model using historical value added of animal husbandry. For the two specific cases as mentioned in our previous manuscript, our modeled aggregate livestock loss for event (a) in 1996 in Yushu and Guoluo prefectures was approximately 1.20 million heads/units (turned from 2.64 million sheep units modeled given the cattle-to-sheep ratio in Qinghai Province), and the historical record was 1.48

million[1] livestock recorded), and for event (b) in 1998 in Naqu Prefecture was 0.72 million head/unit (turned from 1.59 million sheep units), compared to 0.82 million livestock. Therefore, our model result did capture the loss of major events in specific regions, although it still suffered from uncertainty.

In our revision, we will put the validation result into discussion. In addition, model results uncertainty will be added into the limitation section.

Page 21 Line 8 "two critical indices": Is this presented in the Result section?
RE: The "two critical indices" were referring to disaster duration and growing season aggregate precipitation, two variables that critically determines livestock mortality rate in our vulnerability function.

In the revision, we have deleted this part to make it clear. "*A more favorable choice is to adopt an index-based structure using livestock mortality rate as predicted by our model.*"

Page 22 Line 18: How the study can be applied for future? I consider that the method used here is not suitable when the climate is changing.
RE: Thank you for your question. Reviewer 1 has raised similar comment (comment#3 of reviewer#1). So we will refrain from claiming so in the current manuscript and leave the discussion for the next study.

Some discussion might answer your question here. In our modeling process, we have been trying to make our model framework capable of incorporating the changing climate and socioeconomic development. Following the existing work of climate change risk assessment (Tachiiri and Shinoda, 2012; Carleton and Hsiang, 2016; Winsemius *et al.*, 2016; Kinoshita *et al.*, 2018), our model consists of a set of response relationships: 1) a hazard module defines the relationship between daily weather condition and the occurrence (identification) of snow disasters; 2) a vulnerability function defines the relationship between even mortality rate and hazard intensity (duration, wind speed, growing season precipitation) and prevention capacity (as proxied by socioeconomic variable); 3) exposure in terms of herd size is used only in a multiplicative way to derive the final risk metrics in terms of sheep units.

In such a structure, climate condition and socioeconomic condition are merely inputs to our model, rather than a part of the model. Climate change will lead to changes in model input, and correspondingly model output. And that is what we want the model the have the capability to capture: in the short-term future, will the changing climate lead to more or less frequent snow disasters, with shorter or longer duration? Together with the enhanced prevention capacity, will the mortality risk increase or decrease correspondingly? For the first question, our hazard module is capable of identifying/simulating snow disaster days based on climate inputs mimicking meteorological observers' decision using machine learning algorithm: given daily maximum , mean and min temperatures, precipitation, and maximum and mean wind speed, the module can exactly derive corresponding snow disaster event set. Applying it to future climate scenario can then generate future event set and investigate the change of disaster event
* * *
[1] The figure 1.08 million in the previous manuscript was from a literature, but unfortunately it is not consistent with those reported in the books. We have corrected it.

frequency and intensity (duration) in the future. For instance, in a warmer climate we may expect snow disaster events with less frequency and shorter duration.

In summary, climate change will not influence the model structure, but certainly it will change the model results. That is how the climate changing is taken into account in our model.

[Technical corrections]

Page 1 Lines 20,22: 1/20a -> 20 years (also for all similar expressions).

RE: Thanks for pointing this out. We have corrected it through out the manuscript and in the figures.

Pages 14-15 Fig 6: To be multi-colored like Fig 7 would be more reader-friendly.

RE: Thanks for the suggestion. We have updated Fig. 6 to use multi-color in the map.

Page 9 Line 7: "although unlikely" should be rephrased with better expression.

RE: Thanks for the suggestion. We have removed the phrase from the text.

Page 5 Lines 4-6: Hard to understand. Too many "and"s. "its needs" -> "it needs"? Delete one of the two "provide"s?

Page 10 Line 14: Fig. 3 -> Fig. 4?

Page 13 Line 2: topology -> topography?

Page 19 Line 22: There is no Table 2.

Page 19 Line 26 (also in Page 21 Line 28): higher -> longer.

Page 20 Line 8: Fig A2 -> A3?

RE: Above comments are related to typos. We have revised/corrected them as suggested.

**References**

Carleton, T. A. and Hsiang, S. M. (2016) 'Social and economic impacts of climate', *Science*, 353(6304). doi: 10.1126/science.aad9837.

Diffenbaugh, N. S. and Giorgi, F. (2012) 'Climate change hotspots in the CMIP5 global climate model ensemble.', *Climatic change*. Springer, 114(3–4), pp. 813–822. doi: 10.1007/s10584-012-0570-x.

Elith, J., Leathwick, J. R. and Hastie, T. (2008) 'A working guide to boosted regression trees', *Journal of Animal Ecology*. Wiley/Blackwell (10.1111), 77(4), pp. 802–813. doi: 10.1111/j.1365-2656.2008.01390.x.

Gu, H., Yu, Z., Wang, J., Ju, Q., Yang, C. and Fan, C. (2014) 'Climate change hotspots identification in China through the CMIP5 global climate model ensemble', *Advances in Meteorology*, 2014. doi: 10.1155/2014/963196.

GUO, D.-L., SUN, J.-Q. and YU, E.-T. (2018) 'Evaluation of CORDEX regional climate models in simulating temperature and precipitation over the Tibetan Plateau', *Atmospheric and Oceanic Science Letters*. Taylor & Francis, 11(3), pp. 219–227. doi: 10.1080/16742834.2018.1451725.

Hijmans, R. J., Phillips, S., Leathwick, J. R. and Elith, J. (2011) 'Package "dismo"', *R Package*, p. 55. doi: 10.1016/j.jhydrol.2011.07.022.

Kinoshita, Y., Tanoue, M., Watanabe, S. and Hirabayashi, Y. (2018) 'Quantifying the effect of autonomous adaptation to global river flood projections: application to future flood risk assessments', *Environmental Research Letters*. IOP Publishing, 13(1), p. 014006. doi: 10.1088/1748-9326/aa9401.

Kuang, X. and Jiao, J. J. (2016) 'Review on climate change on the Tibetan Plateau during the last half century', *Journal of Geophysical Research: Atmospheres*. Wiley-Blackwell, 121(8), pp. 3979–4007. doi: 10.1002/2015JD024728.

Michel-Kerjan, E., Hochrainer-Stigler, S., Kunreuther, H., Linnerooth-Bayer, J., Mechler, R., Muir-Wood, R., Ranger, N., Vaziri, P. and Young, M. (2013) 'Catastrophe risk models for evaluating disaster risk reduction investments in developing countries.', *Risk analysis : an official publication of the Society for Risk Analysis*, 33(6), pp. 984–99. doi: 10.1111/j.1539-6924.2012.01928.x.

Tachiiri, K. and Shinoda, M. (2012) 'Quantitative risk assessment for future meteorological disasters: Reduced livestock mortality in Mongolia', *Climatic Change*, 113(3–4), pp. 867–882. doi: 10.1007/s10584-011-0365-5.

Wang, W., Liang, T., Huang, X., Feng, Q., Xie, H., Liu, X., Chen, M. and Wang, X. (2013) 'Early warning of snow-caused disasters in pastoral areas on the Tibetan Plateau', *Natural Hazards and Earth System Science*, 13(6), pp. 1411–1425. doi: 10.5194/nhess-13-1411-2013.

Winsemius, H. C., Aerts, J. C. J. H., Van Beek, L. P. H., Bierkens, M. F. P., Bouwman, A., Jongman, B., Kwadijk, J. C. J., Ligtvoet, W., Lucas, P. L., Van Vuuren, D. P. and Ward, P. J. (2016) 'Global drivers of future river flood risk', *Nature Climate Change*, 6(4), pp. 381–385. doi: 10.1038/nclimate2893.

Wu, J., Li, N., Yang, H. and Li, C. (2007) 'Risk evaluation of heavy snow disasters using BP artificial neural network: the case of Xilingol in Inner Mongolia', *Stochastic Environmental Research and Risk Assessment*, 22(6), pp. 719–725. doi: 10.1007/s00477-007-0181-7.

Xin, Y., Du, T., Xin, Y., Wu, A. and Lu, F. (2011) 'The Evaluation of Carrying Capacity of Grassland in Qinghai', *Qinghai Prataculture*, 20(4), pp. 13–22.

---

## Author Response (AR1)

**Response to Editor Decision:** Reconsider after major revisions (further review by editor and referees) (27 Dec 2018) by Margreth Keiler

Dear Prof. Keiler

Thank you very much for your consideration of our paper for the potential publication and your suggestions about the major revision. We have carefully revised the manuscript following comments point-by-point, as well as the revision plan we prepared and submitted earlier. We prepared three documents as requested: (1) a point-to-point reviewer response document including original comments/questions, our response, and corresponding revisions made in the manuscript, (2) a track-change manuscript showing all the detailed modifications in the manuscript, and (3) a clear manuscript after revision.

Again, we appreciate your kind help in the reviewing and revision process. We look forward to further updates from you.

Warm regards,

Tao Ye
On behalf of the co-authors

**Reponses to Reviewer #1**

We greatly appreciate your comments and suggestions. We have revised the manuscript according to what you have pointed out and believe that the quality of this manuscript has been improved as a result of the revision. We have included our detailed responses to each of your comments raised. Your original comments are in bold.

**General comments:**

Natural hazards that lead to disasters can cause tremendous impacts on societies, the environment, and economic wealth of the affected countries. Climate change will exacerbate existing challenges relating to livestock snow disaster risk. Adapting to climate change is a necessity for sensitive areas and those that are vulnerable to climate change such as the Qinghai-Tibetan Plateau. To investigate and better understand the risk of livestock snow disasters in the Qinghai-Tibetan Plateau is critical to towards sustainability of grassland animal husbandry and livelihood of the herdsmen. This topic fits well with the mission and scope of Natural Hazards and Earth System Sciences. However, there are still many flaws in the current manuscript.

**Special comments:**

1. There are many abbreviated symbols in the paper such as QTP, PRA, SDD, BRT etc. Because of too many abbreviations, readers often get confused. It is suggested that a separate symbol page should be set up in front of the manuscript.

RE: Thank you for your kind suggestion. We have added list of abbreviation in the appendix as suggested by the editor. Please refer to the first page of appendix for details.

2. At present, there are many good quantitative methods for the study of snow disaster vulnerability and risk in alpine pastoral areas in particular on the Qinghai-Tibetan Plateau and Inner Mongolia Plateau. In the literature review in the "Introduction", the authors review this issue incompletely. The literature covered is also very limited. And main viewpoint may be biased. For example:

*Page 2, lines 19-20, "The first type employs an ordinal risk assessment framework in which the risk index is derived by integrating several indices representing different components of risk";*
*Page 2, lines 12-13, "The other risk assessment approach is quantitative, often called the probabilistic risk assessment (PRA), in which risk is measured with a probability distribution of socioeconomic losses (consequences)";*
*Page 2, lines 24-32, "However, studies applying PRA to livestock snow disasters have been limited. Bai et al. (2011) published one of the first trials in applying the PRA framework to a livestock snow disaster risk assessment. In their study, winter season (November to April of the preceding year) average daily snow depth was used to describe snow hazard intensity. Physical vulnerability, a function of livestock mortality rate in response to snow depth, was fitted using historical case data. Using annual average snow depth computed from satellite-retrieved data, return-period livestock mortality and mortality rates were derived as the final risk metrics. Based on their method, quantitative livestock snow disaster risks were mapped nationwide in*

*China (Shi, 2011). The major flaw of this method was the mismatch between the event-based vulnerability function and annual measure of snow hazard. In another work focusing on Mongolia, a vulnerability function trained from a tree-based model was used, but still on an annual basis".*

RE: Thank you for your kind suggestion. We did another round of careful literature search and review and found several important articles i.e. (Dong and Sherman, 2015; Miao et al., 2016; Nandintsetseg et al., 2018; Wei et al., 2017; Yeh et al., 2014). We have re-organized our literature review and added these references into the review section following your suggestion and suggestions from Reviewer #2 (comments 1 and 2). Please refer to page 2, lines 5 – page 3 line 5 in the revised clear manuscript for details (or page 2 line 5 – page 3 line 25 in the manuscript with track changes).

References related to this comment:

Dong, S. and Sherman, R.: Enhancing the resilience of coupled human and natural systems of alpine rangelands on the Qinghai-Tibetan Plateau, Rangel. J., 37(1), i–iii, doi:10.1071/RJ14117, 2015.

Miao, L., Fraser, R., Sun, Z., Sneath, D., He, B. and Cui, X.: Climate impact on vegetation and animal husbandry on the Mongolian plateau: a comparative analysis, Nat. Hazards, 80(2), 727–739, doi:10.1007/s11069-015-1992-3, 2016.

Wei, Y., Wang, S., Fang, Y. and Nawaz, Z.: Integrated assessment on the vulnerability of animal husbandry to snow disasters under climate change in the Qinghai-Tibetan Plateau, Glob. Planet. Change, 157(March), 139–152, doi:10.1016/j.gloplacha.2017.08.017, 2017.

Yeh, E. T., Nyima, Y., Hopping, K. A. and Klein, J. A.: Tibetan Pastoralists' Vulnerability to Climate Change: A Political Ecology Analysis of Snowstorm Coping Capacity, Hum. Ecol., 42(1), 61–74, doi:10.1007/s10745-013-9625-5, 2014.

3. Page 3, lines 13-15, "*Worldwide, the QTP suffers from some of the highest livestock snow disasters due to its large area of snow cover area, longlasting snow cover days, and nomadic grazing. This region is also a hot spot in climate change. Quantitative risk assessments for the present day will likely be a significant source of information for disaster risk reduction*". The above sentence should be moved to before line 5 on the second page.

Delete lines 15-16 of the second page, "*In addition, the framework can be adapted for livestock mortality in snow disasters in the context of future climate change analysis, and therefore support climate adaptation planning for local government and herding communities*".

RE: These places has been revised according to your suggestion.

4. In the "Materials and Methods" section, the Qinghai-Tibet Plateau as case area, it is necessary to have a more comprehensive description of the geographical, environmental, social, and economic backgrounds of the QTP, especially the role of livestock in livelihood for local people.

RE: Per your suggestion, we have added a sub-section exclusively introduce the QTP, including its geographical, environmental social and economic backgrounds, with emphasis on the role of livestock in livelihood for local people. Please refer to section 2.1 in the revised manuscript for details.

5. We know the positive intervention of humans on the grassland ecosystem and that the grassland carrying capacity could be elevated with a reduction of harmful human activities (adverse effect), an increase of disaster prevention capacity. For example, the proportion of

fenced pasture area to the total usable grassland (to show the capacity of grassland biomass to regenerate), the warm shed area per unit of livestock (to illustrate the capacity of livestock to prevent freezing disasters) and the proportion of sown grassland area to the total usable grassland (to descript the capacity of balancing forage supply and demand), accessibility of traffic and information (to depict the capacity of disaster response or prevention), if the above key factors are missing, in other words, if the authors do not emphasize the socio-system intervention for livestock snow disaster assessment, it will be very difficult to objectively assess the risk of snowstorms in livestock.

6. Page 4, line 10, "*prevention capacity as measured by gross domestic production (GDP) of the underlying county*", GDP as prevention capacity is not a scientific proxy, indeed, local fiscal revenue and the intensity of infrastructure construction in animal husbandry (including alpine grassland) are the key to reducing vulnerability and risk of livestock snow disaster.

7. Similarly, page 8-9, in "2.4 Loss modelling", as one of loss index, GDP at county level is not consistent with the risk topic of livestock snow disaster. It is suggested that the added value of animal husbandry at county level should be adopted.

RE: Above comments (5, 6, and 7) are related to each other and are responded together.

We totally agree that it needs a thorough understanding of vulnerability to snow disaster before a good risk assessment carries out. You have offered important insights into local herders' coping capacity to snow disasters, and the factors that you mentioned (fenced pasture area, warm shed area, sown grassland, and accessibility) are critical in deciding vulnerability to snow disaster. These variables are valuable, but mostly only available for certain regions and can only be obtained from interview/survey. Per your suggestion, we checked again the statistical yearbooks, including the provincial statistical yearbooks of Qinghai and Tibet which date back to 1989, and the National County (City) Socioeconomic Statistical Yearbook (2000~ ), but found little data such as fenced pastures, warm shed areas, and sown grassland area.

So we kept the strategy of using a proxy variable to indicate prevention capacity. Following your suggestion, we collected data on "fiscal revenue" (*Fiscal_Rev*) and "added value in animal husbandry" (*Value_Add*), which could be the first-best choices to denote prevention capacity. In addition, we also considered Fiscal Expenditure (*Fiscal_Exp*) and GDP per capita (*GDP_PC*). All these values were turned to 2015 Yuan. These variables were slightly-to-moderately correlated with GDP. The Pearson correlation coefficients between *Value_Add*, *Fiscal_Rev*, *Fiscal_Exp*, *GDP_PC* and *GDP* are: 0.336, 0.760, 0.420 and 0.223, respectively.

We re-ran our generalized additive model (GAM) as shown in Eq (1) by replacing GDP with each of the four variables, and performed model diagnostics to check goodness-of-fit as well as response curves. Summary statistics of model runs are provided as below:

Table R1 Generalized additive model results by using different socioeconomic factors indicating prevention capacity

| $\ln LR$ $= s(Duration)$ $+ s(Wind) + s(P) +$ | R-sq.(adj) | Deviance explained | GCV | N(sample) | Significance level of the socioeconomic factor |
|---|---|---|---|---|---|
| $s(GDP)$ | 0.554 | 62.1% | 2.5105 | 79 | At 0.01 level |
| $s(Value\_Add)$ | 0.563 | 62.5% | 2.5508 | 73 | At 0.01 level |
| $s(Fiscal\_Rev)$ | 0.516 | 58.4% | 2.8392 | 73 | Not significant |

| | | | | | |
|---|---|---|---|---|---|
| s(Fiscal_Exp) | 0.524 | 58.4% | 2.7301 | 73 | At 0.01 level |
| s(GDP_PC) | 0.506 | 57.2% | 2.8561 | 74 | Not significant |

As shown in the table, variable *Fiscal_Rev* cannot improve the prediction of ln*LR* (natural logarithm of mortality rate). *Value_Add* is capable of deriving competing results. The response curves of the variables are also similar: ln*LR* showed downward slope with each of the three variables, indicating decreasing loss rate in response to enhanced prevention capacity.

Given above analysis, together with your suggestion, we have decided to use value added of animal husbandry in our vulnerability function, and update all our results throughout the manuscript. Specifically:

1) We have updated the texts describing the vulnerability function
2) We have re-calculated event and annual loss (mortality rate and mortality), and updated the result maps and tables

In the revised manuscript, we have re-organized our description on the vulnerability function to address the changes made according to your suggestion and our additional analysis listed above. Please refer to section "*2.2.2 Vulnerability function*" in the revised manuscript for details. In addition, we have also supplied a supplementary material about the GAM fitting details of the vulnerability function we have used.

8. Page 7, lines 3-6, the authors stated that "*Historical snow disaster event data with the time of each event for each meteorological station were used to train the BRT model. These data were obtained from two sources. Records for 1980–2007 were obtained from W. Wang et al. (2013) while records from 2008–2015 were obtained from the China Meteorological Science Data Sharing Service System (CMSDS, http://data.cma.gov.cn)*." However, are the identification criteria of the two snowstorm records sources consistent?

RE: This comment is high related to Reviewer #2's comment (Page 7 Lines 4-6: Are there no bias between the two data?). According to (Wang et al., 2013), the data for period of 1980-2007 were obtained from the yearbooks of meteorological disasters. Therefore, these data were originally recorded by provincial meteorological administrations officially, and published as a collection in yearbooks. The data for 2008-2015 were directly obtained from China Meteorological Administration in digital format. Therefore, they are both from official records from meteorological administration, the standards in identifying snow disasters are the same according to local Meteorological Administration officials.

In the revision, we have added the information to clarify that the potenail bias between two data is very limited. "*Records for 1980–2007 were a collection of snow disaster records published in 6 provincial meteorological yearbooks neighboring the Plateau (Wang et al., 2013b). Records from 2008–2015 were obtained from the China Meteorological Science Data Sharing Service System (CMSDS, http://data.cma.gov.cn). Records in both datasets are official observations by the meteorological administrations and are consistent with each other in terms of observation standards.*" Please refer to page 6, lines 18-22 in the revised manuscript for details (or page 8 lines 6-11in the manuscript with track changes).

9. Page 8, in "2.3 Exposure", the herd size as a critical proxy of exposure, although the spatial

distribution of livestock size can reflect the extent of snowstorm exposure of livestock, it is well known that the Qinghai-Tibetan Plateau has a vast area with obvious spatial differences, and the distribution density of livestock (the number of livestock per unit area) may be more scientifically and accurately describe the spatial feature of snowstorm exposure.

RE: Thank you for your suggestion. We have changed our term from "herd size" to "herd density" where exposure is discussed. In the exposure map we derived (Fig. A2), the unit has already been (Sheep unit/ha). Accordingly, we will update the risk metrics map in terms of mortality (Fig.7) to show the loss measured with sheep units/km$^2$.

10. Page 9, lines 1-4, I don't understand that *"County level GDP values were assigned to each grid within its boundary. We used constant GDP values for 2015 for two reasons. First, the results can be directly treated as a stationary time series for estimating the probability distribution, as the influence of prevention capacity improvement has been removed. Second, it meets the goal of risk assessment, to estimate the likelihood of potential loss in the near future"*. The GDP of each county changes with time. Dynamic GDP should be used instead of static GDP to predict the probability of loss, which is not consistent with reality.

RE: Thanks for your comment. In our vulnerability function, GDP (has been changed to value added of animal husbandry per your comments 5-7) was used as an indicator of prevention capacity. Therefore, whether to use historical dynamic value or a static present value essentially depends on our purpose of analysis.

    1) If we are modeling actual historical losses for model calibration and verification purposes, historical dynamic value should be used (for such discussion, please refer to the response to reviewer#2's comment regarding Page 19 Lines 15-16 and Page 19 Lines 27-28).

    2) If we are assessing livestock risk (the probability of potential loss) for the next couple of years, then using present-day prevention capacity would be a better choice than using historical prevention capacity. Then, our risk assessment effort tries to answer "if historical events occur in nowadays prevention capacity, how would the probability distribution of the loss will be". Correspondingly, the risk metrics would be meaningful for prevention planning and insurance implications because we are considering the near future given today's situation.

    Technically, to fit a probability distribution from samples of loss requires that the sample data must be at least stationary in its mean and variance, so as to remove any technical, environment, or prevention capacity change effects. This is the reason in many risk assessment research, historical loss must be "detrended" before it was fit (Lobell and Burke, 2010; Maddala, 1977; Ye et al., 2015). In our case, as GDP (or value added in animal husbandry) keeps growing along the time, modeled losses based on historical dynamic GDP (or value added of animal husbandry) will contain obvious trend and therefore cannot be used directly to fit any probability distribution.

    In order to better explain the difference, we modeled annual winter losses (from September ~ June of the next year) using both historical (dynamic) and static (2015) value added of animal husbandry, and the time series are shown below (Figure R1).

[Figure]

Figure R1 Modeled total livestock loss over the Tibetan Plateau. The blue time series was losses modeled using constant *Value_Add* of year 2015, assuming present-day prevention capacity (static). The orange time series was losses modeled using *Value_Add* of historical values (dynamic), assuming historical prevention capacity. All the losses are for a specific snow disaster season from September to the next June rather than a civil year.

Loss modeled using historical value added of animal husbandry (*Value_Add,* the orange time series in Figure R1) obvious contained trends. It showed an obvious downward trend. Fitting a probability distribution would over-estimate the size of risk. Using a static present value of one recent year, it technically meets the requirement of a stationary time series (the blue trend line is flat in Figure R1). Then losses derived from our event-based model are derived with the prevention level of historical period. Their difference demonstrates the effect of improved prevention capacity (which might echoes the Comment #1 of reviewer2, and provides meaningful discussion for your Comment #11).

In the revision, we have explicitly describe our modeling efforts of modelling annual aggregate losses by using both historical dynamic and 2015-static value added of animal husbandry in "*2.2.4 Loss modelling*". Correspondingly, we have also supplied corresponding results as shown in Figure R1 in the revised manuscript (Figure 6).

11. The discussion part only deals with the content of spatial pattern (4.1 Spatial patterns of livestock snow disaster risk in the QTP). As an important part of risk change, the characteristics of dynamic and temporal variation cannot be absent. Moreover, the authors do not pay attention to the causal relationship between risk and its influencing factors.

RE: Thank you for your comment. We added a new subsection in the result section to show temporal changes of livestock mortality derived from our model, and enrich our discussion by comparing it to historical losses observed. Please refer to the new section "3.1.2 Model-derived annual snow disaster loss, 1980–2015" for details.

The discussion over the causal relationship between risk and its influencing factors can be partly be done by comparing the dynamics of the modeled loss with historical prevention capacity and static present-day prevention capacity, as shown in Figure R1. We have also added discussion about the role of improved prevention capacity in a new section "*4.2 Temporal changes of livestock snow disaster loss and its drivers*".

12. In "4.3 Risk-informed implications" *Page 20, line 22, "Our results imply that the present level of preparedness in local regions are far from sufficient";*
*Page 21, line 5, "Due to the difficulty in improving prevention capacity, insurance schemes are*

*needed to provide relief";*

From the perspective of above mentioned sentences, this is not the inspiration of the article analysis, but the main existing problems.

RE: Thank you for your comment. We have revised this part by merging this section into section "*4.3 Advantages of the event-based PRA*". We have deleted the general issues and main existing problems, and tried to concentrate on the unique quantitative information that are offered by the event-based PRA framework, and its implication in risk-informed risk reduction actions.

13. The language of the manuscript is rather deficient and requires the re-editing of native speakers.

RE: We have sent the paper for professional proof-reading service. A proof of the proof-reading service has been included in the re-submission.

**Reponses to Reviewer #2**

We greatly appreciate your comments and suggestions. We have make a careful plan to revise the manuscript according to what you have pointed out and believe that the quality of this manuscript will be improved as a result of the revision. We have included our detailed responses to each of your comments raised. Your original comments are in bold.

**[General comments]**

"Probabilistic risk assessment of livestock snow disasters in the Qinghai-Tibetan Plateau" by Ye et al. applies boosted regression tree and general additive modeling methods to the snow disasters in the Qinghai-Tibetan Plateau, as an event-based evaluation, and the results are basically consistent with existing studies. The research topic is within the scope of the journal, but there are some substantial flaws in the study, which should be addressed. The major concerns are:

1: The advantage of event-based evaluation is not clear. Rather than hay preparation based on event-based analysis of this study, that based on annual-based analysis will work well when the intervals between the two events are very short (as time to prepare for the next event is not enough, preparation for annual basis is better, particularly when modeled annual frequency > 1).
Page 19 Sect. 4.2: As "more than one snow disaster a year is unlikely", annual evaluation is enough, isn't it? For me it looks to prepare hay based on annual evaluation is OK.
2. The authors say probabilistic analysis is one of the advantages of the study. However I consider a year-by-year evaluation is more sophisticated, and the probabilistic analysis used here is not necessarily an advantage, but a result of taking a simple evaluation dealing with what should be separately treated as one set of data. An effective PRA would be a result from a probabilistic function, not a result from treating various conditions as one case.

RE: Above questions are inter-related and are therefore responded together.

We totally agree that annual based modeling has its own advantages, but the event-based approach is also very important. By nature, livestock snow disaster can occur multiple times in a winter, and the losses would accumulate event by event. Capturing the details of event occurrence and intensity are important for following aspects:

1) ln$LR$ (natural logarithm of loss rate) shows a concave relationship with disaster duration (can be found later in details in response to your comment #4). Therefore, the total loss of one event lasting 30 days and two-event in one winter lasting 15 days each will be different. Knowing only the aggregate duration in a year cannot reflect such important details.

2) Although presently our data shows that historically it has been less likely to have more than 1 snow disaster in a winter for most parts of the Qinghai-Tibet. But if we are considering a method that can be generalized, then we need to consider the possibility of changing frequency and intensity, and the capacity to model it, particularly with the observed and projected slight increase in precipitation on the Plateau (GUO et al., 2018; Kuang and Jiao, 2016). Using the annual loss approach cannot capture such changes.

3) From the perspective of risk-informed actions, annual evaluation (i.e. potential aggregate duration) would be temporarily good for preparedness by the government and community. But it can hardly work for risk-transfer mechanisms as insurance schemes were mostly based on events. This is also the critical reason that catastrophe risk models are mostly built on event-basis (Michel-Kerjan et al., 2013).

For the advantage of this study, we totally agree with you that a year-by-year evaluation is more sophisticated (than a simple statistical analysis). So the advantage should be event-by-event loss modeling, which is even more sophisticated than year-by-year evaluation, rathern than probabilistic analysis. Benefited from your comment, we have added these discussions into the introduction section, page 2 line 29 – page 3 line 4 (or Page 3 Lines 15-25 in the track-change version) and discussion section (section "*4.3 Advantages of the event-based PRA*") to better highlight the advantage of event-based modeling instead of probabilistic analysis.

3: To evaluate livestock number by carrying capacity, and to evaluate carrying capacity by grassland type is questionable. The appropriateness of them should be more carefully discussed.

Page 8 Lines 18-20: More careful discussion on the validity of this method is needed. I want to see the scatterplot of observed and estimated values.

Page 8 Lines 21-22: More careful discussion on the validity of this method is needed. I want to see the scatterplot of observed and estimated values.

Page 21 Lines 19-20: More careful evaluation is needed for the performance of herd size estimation.

RE: Thank you very much for your comment. Above comments are related and therefore responded together.

1) The appropriateness of evaluating livestock number by carrying capacity

We first need to clarify that we are trying to use carrying capacity to evaluate livestock number EXPOSED to snow disaster, but not the total livestock number. In the study area, only livestock grazing on grassland in pastoral and agro-pastoral regions (central to western part of Tibet Plateau) are exposed to snow disaster. Livestock kept in ranches or industrial livestock farms in agricultural regions (mostly the eastern and lower altitude parts of the Plateau) are not. When estimating exposure, we were trying to estimate the livestock number grazing on grassland.

The livestock number grazing on grassland is essentially determined by livestock carrying capacity, which exactly defines the maximum livestock number that can graze in a specific area of grassland, given local forage productivity and grazing style. Since 2011, the Tibet Autonomous Region started the program of "subsidy and award policies for grassland ecological conservation efforts" to reduce the number of livestock and conserve grassland. In 2014-15, news articles intensively reported that "Tibet has basically reached forage-livestock balance" (government release, http://www.gov.cn/xinwen/2014-05/12/content_2677946.htm, in Chinese). In other words, the actual livestock number grazing on grassland (and exposed to snow disaster) is highly consistent to the carrying capacity officially designated.

2) The appropriateness of evaluating carrying capacity by grassland type

Evaluating carrying capacity by grassland type is the way that adopted by both official calculation technical manual (*Ministry Standard of Calculation of Rangeland Carrying Capacity issued by Ministry of Agriculture of China*, NY/T 635-2015) and academic studies regarding forage-livestock balance, i.e. (Xin et al., 2011). Grassland type essentially determines

the key values of evaluating carrying capacity, i.e. forage regrowth percentage, proper utilization rate of rangeland (of different grazing seasons), conversion coefficient of standard hay. Using grassland type as an identifier of different carrying capacity is also the widely adopted way of presenting the evaluation results.

3)  Verification of livestock number estimated

We obtained official release of carrying capacity by Tibet Autonomous Region from *Statistical Materials of Grassland Resource and Ecology of Tibet Autonomous Region* published by the Department of Agricultural and Pastoral of Tibet Autonomous Region in year 2011. We used zonal statistics to derive county-level carrying capacity estimated by our method, and compared with the official release (Figure R2).

[Figure]

Figure R2 Estimated and officially released livestock carrying capacity in Tibet Autonomous Region

As the above Figure R2 shows, the estimated and officially released livestock carrying capacity basically agree with each other.

In the revision, we have 1) clarified that we are estimating the herd size exposed to snow disaster (livestock number on free grazing) rather than total herd size; and 2) further explained the appropriateness of estimating livestock number of carrying capacity, and estimating carrying capacity by grassland type. Please refer to section "*2.2.3 Exposure*" for more details of the revision.

4. Explanation and discussion on Eq. 1 is not sufficient. The form of functions for each term in the right hand side, and the performance of the equation should be clearly presented.
Page 4 Line 10 (eq 1): Present the detailed forms of s(Duration), s(Wind), s(P) and s(GDP). (all parameters of spline curves)
Page 4 Line 14: More detailed performance check is needed (not for lnLR, but for LR). How large is RMSE? Is the error random or systematic? I want to see the scatterplot of observed and modelled values.

RE: Thank you for these questions and comments. In light of the suggestion from Reviewer #1 (please refer to Reviewer#1 Comments 5,6, and 7), we have updated the model by using value added of animal husbandry of the underlying county instead of GDP as the indicator of prevention capacity. Correspondingly, Eq(1) has been updated to: $\ln LR = s(Duration) + s(Wind) + s(P) + s(Value\_Add)$. Details of the updated model are provided below.

(1) Model-fitting statistic:

```
Family: gaussian
Link function: identity

Formula:
lnLR ~ s(Duration) + s(Value_Add, k = 4) + s(maxWind) + s(P)

Parametric coefficients:
            Estimate Std. Error t value Pr(>|t|)
(Intercept)   0.5300     0.1708   3.103  0.00287 **
* * *
Signif. codes:  0 '***' 0.001 '**' 0.01 '*' 0.05 '.' 0.1 ' ' 1

Approximate significance of smooth terms:
               edf Ref.df     F  p-value
s(Duration)  5.626  6.705 9.706 1.96e-08 ***
s(Value_Add) 1.000  1.000 8.878  0.00407 **
s(maxWind)   2.732  3.440 2.747  0.04463 *
s(P)         1.000  1.000 2.809  0.09864 .
* * *
Signif. codes:  0 '***' 0.001 '**' 0.01 '*' 0.05 '.' 0.1 ' ' 1

R-sq.(adj) =  0.563   Deviance explained = 62.5%
GCV = 2.5508  Scale est. = 2.1593    n = 74
```

Figure R3 Fitting statistic of the updated GAM model

(2) The response curves (spline curves)

[Figure]

Figure R4 Response curves of the updated GAM model

Its response curves (Figure R4) indicate that: (1) ln*LR* is increasing with snow disaster duration. Duration up to 15-18 d is a critical period that mortality will increase rapidly. (2) ln*LR* decreases with value added of animal husbandry (*Value_Add*), indicating the effect of stronger prevention capacity in reducing mortality, i.e. government expenditure in reserving hay for preparedness, and subsidy to herders to build/enlarge warm sheds. (3) An inverted-U shaped relationship between daily maximum wind speed and ln*LR*. The up-slope part indicates the increasing stress of stronger wind on livestock, but the down-slope part (beyond 5-6 m/s)

indicate herder's reaction to stop free-grazing and keep herds in shelters (Wu et al., 2007). (4) ln*LR* decreases with growing season precipitation. Larger growing season precipitation indicates more abundant food for livestock in summer, and therefore better body-condition in resisting low temperature and lack of food in snow disaster times.

(3) Model performance diagnostics

We performed 10-fold cross validation, and found RMSE, MAE and ME for the model were 1.747, 1.325, and -0.002, respectively.

The performance diagnostics charts of the model (Figure R5) indicate that 1) QQ plot is very close to a straight line, suggesting our distributional assumption of normality about ln*LR* is reasonable. 2) The variance is approximately constant as the mean increases. 3) The histogram of residuals appears approximately consistent with normality. 4) the response against fitted values show a positive linear relationship in the scatter plot.

[Figure]

Figure R5 Performance diagnostics charts for the updated GAM model

In the revision, we have supplied more details about the model fitted in to the manuscript. Please refer to section "*2.2.2 Vulnerability function*" for more details. In addition, model details shown here, including the fitting statistics, response curves, and performance diagnostics charts (Figure R3~Figure R5) has been provided as a supplementary material for reference.

[Specific comments]

Page 3 Line 1: Do you mean that the final metrics should always be mortality, not mortality rate? It looks the study considered mortality rate as the final metrics, and I think each study can use its own final metric.

RE: Thank you for your comment. The sentence was misleading. In the revision, we have deleted the latter part of this sentence and kept "*Compared to earlier works, they successfully extended the framework to future climate change analysis*" only.

Page 3 Line 13: Provide the rationale of "some of the highest livestock snow disasters". What is "highest", by the way? Largest damage? Highest frequency?

RE: We intended to say "one of the regions with the highest risk" in terms of both high frequency and large damage. In the revision, we revised this sentence to "*Worldwide, the QTP is a region that has most-suffered from livestock snow disasters due to its large snow cover area, long-lasting snow cover days, and nomadic grazing (Li et al., 2018).*" (page 3, lines 11-12 in the clear version)

Page 3 Line 14: Provide literature for "This region is also a hot spot in climate change".
RE: References will be provided into the text as suggested in the revision, i.e. (Diffenbaugh and Giorgi, 2012; Gu et al., 2014).
Diffenbaugh, N. S. and Giorgi, F.: Climate change hotspots in the CMIP5 global climate model ensemble., Clim. Change, 114(3–4), 813–822, doi:10.1007/s10584-012-0570-x, 2012.
Gu, H., Yu, Z., Wang, J., Ju, Q., Yang, C. and Fan, C.: Climate change hotspots identification in China through the CMIP5 global climate model ensemble, Adv. Meteorol., 2014, doi:10.1155/2014/963196, 2014.

Page 4 Fig 1: It is difficult to understand the relationship between Timing, and Duration and Wind speed from the figure.
Page 4 Sect. 2.1: Fig 1 indicates GDP part is the vulnerability function, while Sect. 2.1 reads Eq. 1 (to derive mortality rate from GDP as well as hazard indicators). Which is true?
RE: Above questions are related and are therefore responded together.

The figure presented in the online document was not complete due to unknown technical reasons when generating .pdf files. The correct figure should be:

[Figure]

Figure R6 Corrected technical flow of the study (Fig 1 in the manuscript)

In the modeling process, we derive timing (both starting and ending dates) from the "snow disaster event identifier/ simulator". Then the duration of the event (days between the starting

and ending dates) and the wind speed during this period can be derived (as we have daily wind speed data).

GDP (has been replaced by value added of animal husbandry) was one out of the four inputs of the vulnerability function (others were hazard indicators), which was used to indicate prevention capacity. In the revision, we have corrected and updated Fig. 1 in the manuscript by moving the vulnerability function into box "Event loss rate modeling" to keep it consistent with Eq. 1.

Page 5 Line 19: How the data of the previous days are used?

Page 6 Line 14: Why since the last snow fall day?

RE: The two comments are related and are therefore responded together.

We used average daily precipitation (snowfall) since the last snowfall day (the last day with effective precipitation) to represent the diminishing impact/pressure from snowfall as time elapses. In general, the larger amount of snowfall, the longer the impact of the snowfall would be given identical temperature conditions – it would take longer time to thaw. Similarly, the closer a day to the snowfall day, the more likely it would be identified as a "snow disaster" day. It worked especially for deciding whether two snow fall days should be regarded as parts of one single event or two independent events.

Page 6 Lines 12-13: How do you compromise when the two standards are different with each other?

RE: Thank you for your comment. The two standards, although designed for different regions, are not conflicting with each other on the key indicators (Table R2), but shares similar indicators (i.e. continuous days of perpetual snow cover, or number of consecutive snow fall days since the last snow fall day).

Table R2 Variables considered in national and industrial standards for snow disaster

| Snow Disaster Grades in Grazing Regions of China (GB/T20482-2017) | Meteorological Grades of Urban Snow Hazards (QX/T 178-2013) |
| --- | --- |
| Snow depth (cm) | Cumulative snowfall (mm) |
| Grass height (cm) | Maximum daily snowfall (mm) |
| Continuous days of snow cover (d) | Snow depth (cm) |
| % of grassland covered by snow (%) | Number of consecutive snowfall days (since the last snowfall day) |
| | Daily lowest temperature (degree C) |
| | Windspeed (m/s) |
| | Minimum relative humidity *(%) |

We have also clarified this point in the revised manuscript (page 6 line 10 in the clear version, or page 7 line 13 in the track-change version).

Page 7 Lines 1-2: This sentence dose not explain why satellite is not used here.

RE: Thank you for your comment. Data need from satellite imagery is mainly the daily snow cover ("yes/no" data) and snow cover rate (% area covered by snow). One of the major purpose of this study is to develop a probabilistic risk assessment framework for risk assessment future climate change scenarios. For such a purpose, our input variables used in the framework must

be available in climate projections. As far as we know, there are no projections of daily snow cover for the future. This is one of the main reasons that we did not consider it.

In the revision, we have tried to make this point clear to the readers. "*Daily snow cover data were not considered as they were absent in future climate projections. Because our goal is to develop a model framework that can assess both present-day and future risk with climate projections, we refrained from using variables absent in future climate projections.*" Please refer to page 6 lines 13-16 (or page 8, lines 1-5 in the track-change version) for details.

Page 7 Lines 4-6: Are there no bias between the two data?

RE: A similar question has been raised by Reviewer #1 (comment #8). According to (Wang et al., 2013), the data for period of 1980-2007 were obtained from the yearbooks of meteorological disasters. Therefore, these data were originally recorded officially by provincial meteorological administrations, and published as a collection in books. The data for 2008-2015 were directly obtained form China Meteorological Administration in digital format. Therefore they are both from official records from meteorological administration, and the standards in identifying snow disasters are the same. However, we cannot perform a bias check as the data from two different sources do not share any overlapping period.

In the revision, we have added information to clarify that the potenail bias between two data is very limited. "*Records for 1980–2007 were a collection of snow disaster records published in 6 provincial meteorological yearbooks neighboring the Plateau (Wang et al., 2013b). Records from 2008–2015 were obtained from the China Meteorological Science Data Sharing Service System (CMSDS, http://data.cma.gov.cn). Records in both datasets are official observations by the meteorological administrations and are consistent with each other in terms of observation standards.*" Please refer to page 6, lines 18-22 in the revised manuscript for details (or page 8 lines 6-11in the manuscript with track changes).

Page 7 Line 11: Explain the meanings of lr and tc.

RE: Boosted Regression Trees have two important parameters that need to be specified by the user. Tree complexity (*tc*) controls the number of splits in each tree. A *tc* value of 1 results in trees with only 1 split, and means that the model does not take into account interactions between environmental variables. A *tc* value of 2 results in two splits and so on. Learning rate (*lr*) determines the contribution of each tree to the growing model. As small value of *lr* results in many trees to be built. These two parameters together determine the number of trees that is required for optimal prediction. The aim is to find the combination of parameters that results in the minimum error for predictions.

In the revision, we have supplied these information into the text (page 6 lines 26-28, or page 8 lines 15-17 in the track-change version).

Page 7 Lines 13-14: How the number of the variables and prediction power is weighted? Any kind of criteria like AIC or BIC is used?
Page 10 Line 4: Why SD, minWind and Pre were excluded?

RE: SD, minWind and Pre were excluded as they were least important in explaining the response variable. BRT uses a process of variable selection analogous to backward selection in regression. It drops the least important (in terms of relative influence) predictor, then re-fits the model and sequentially repeating the process (Elith et al., 2008). In each step, after the removal of one predictor, the change in predictive deviance is computed relative to that obtained when

using all predictors. Finally, a list containing the mean change in deviance and its standard error as a function of the number of variables removed will be returned (Hijmans et al., 2011). From the list, the optimal number of variables to drop can be identified, i.e. the number of variables that yield the minimum predictive deviance.

In the revision, we have supplied these information about the mechanism of variable selection and model simplification. Please refer to page 6 lines 31-34 (or page 8 lines 20-24 in the track-change version) for details.

Page 7 Line 14: The cross validation is how many fold?
RE: We used a 10-fold cross-validation. This information has been supplied in the revision (page 6 line 34).

Page 7 Line 20: Is "prediction error" random? If they are systematic, to take average may not be a good solution.
RE: The CV estimates of prediction error (predictive deviance in BRT models, Elith et al. 2008) of our BRT model indicates that the error is random.

Page 9 Line 9: How good is the performance of the equation?
RE: This equation is derived using logic reasoning: if a share of livestock died in one event, then the actual herd size exposed to the next event should be reduced correspondingly. As we rarely have two or more subsequent events in one year and one place in our historical records, we cannot evaluate the performance of this equation. We can only say that it is logically correct.

Page 10 Line 7: How relative contribution is calculated?
RE: In BRT, the relative importance is calculated based on the number of times a variable is selected for splitting, weighted by the squared improvement to the model as a result of each split, and averaged over all trees. The relative importance for each variable is scaled so that the sum adds to 100 (Elith et al., 2008). We have added this information as a footnote (page 11).

Page 10 Line 19 "well captured": p-values of 0.118 and 0.189 are not necessarily good (not statistically significant with 10% level). To check the model's representability more carefully, let us see not cumulative but probability density function before accumulation.
RE: We agree with your that we may not use "well captured". But we want to double confirm about the meaning of "not statistically significant with 10% level". The null hypothesis of the Two-sample Kolmogorov–Smirnov test is: "The two samples come from a common distribution", and the alternative hypothesis is: "The two samples do not come from a common distribution" (https://www.itl.nist.gov/div898/software/dataplot/refman1/auxillar/ks2samp.htm). Our test statistics indicated "not statistically significant at 10% level", and therefore we failed to reject the null hypothesis and had to believe that the two samples (observed and predicted) were from a common distribution. It said that our prediction had captured the statistical feature of the observed duration (both event and annual). We posted the probability densities of these variables below, and actually they showed similar thing with the cumulative density function:

[Figure]

Figure R7 probability densities of observed and predicted snow disaster durations

In the revision, we have changed "well captured" to "captured" following your suggestion.

Page 12 Fig. 5: Better to present the annual total number of SDDs.

Page 13 Line 13: Fig 5b shows annual aggregate snow disaster duration? But the caption says "mean event duration".

RE: Two questions above are related and responded together.

We believe that you have been discussing Fig. 5 (b). In the revision, we have followed your suggestion and present the mean annual aggregate duration (mean annual total number of SDDs). Correspondingly, its corresponding description has been updated to keep consistency with the figure (page 12 line 13-18).

Page 13 Line 11 (also for Page 18 Line 1): It is obvious when the Gaussian approximation is used.

Page 13 Lines 18-19: If "The distribution of annual average mortality rate is extremely positively skewed", the Gaussian kernel function (Page 9 lines 21-22) is not appropriate, is it? BTW, is it related to the dependent variable in Eq .1 is lnLR, not LR?

RE: Thank you for your question. The two comments are related and are responded together. There are three points to clarify:

1) We did not use Gaussian approximation over the simulated annual loss rates, which is a parametric and symmetric distribution function. Instead, we used a non-parametric approach called kernel density function (Page 9 lines 21-24) with a Gaussian kernel function (Silverman 1986). The kernel density approach does not specify any functional form of the entire distribution and therefore is flexible to capture probability densities of different degrees of skewness. A simple explanation about the method can also be found in Ker and Goodwin (2000, American Journal of Agricultural Economics).

2) Page 13 Lines 18-19 was describing the distribution of annual average mortality rate of different grids (spatial locations), but not the distribution of annual mortality rate for any specific location. "Extremely positively skewed" means that the grids/regions of high mortality risk take only a small portion of all places on the Plateau, but their annual average

mortality rates were much higher than those of other grids. It is not related to the distributional assumption of dependent variable LR.

3) LR is also positively skewed. Only its natural logarithm (lnLR) exhibits normality.

Page 18 Table 1: What is the trend of actual herd size in QTP? To consider a static herd size is reasonable?

RE: The trend of herd size as a total of Qinghai and Tibet is provided below in Figure R8. As shown in the figure, number of cattle has not changed much for both regions. Number of sheep in Qinghai has not changed much since 2006, but that of Tibet has been decreasing for recent years, mainly due to the forage-livestock balance policy. In terms of aggregate size (1 cattle = 5 sheep units), Qinghai remains quite stable since 2006, and Tibet keeps dropping rapidly up to year 2014. Once it drops to the upper limit of carrying-capacity, it is very likely to keep nearly constant (please also refer to the discussion on your comment #3).

[Figure]

Figure R8 Change of herd size summed from Qinghai and Tibet

In our results, we have added discussion regarding the use of static exposure into the limitation section. Please refer to page 23 lines 23-26 for details (or page 32 line 33 – page 33 line 4 in the track-change version).

Page 18 Line 8: Why mortality becomes small by the constraint of herd size by carrying capacity?

RE: This sentence was misleading. We intended to say that the mortality (sheep units) was small (mostly below 10 per grid), and the main reason was that the estimated carrying capacity was small. In the revision, we will change this sentence to: "*Mortality appears small in Fig. 8, generally several sheep units/km$^2$. However, when aggregated at the prefecture level, mortality remained considerable (Table 1).*" (page 20 lines 4-5, clear version)

Page 19 Lines 15-16: It is better to compare the modeled mortality with observed (historical) ones.

Page 19 Lines 27-28: Is there no possibility that this study overestimates?

RE: The two questions above are related to each other and are responded together.

To further test the performance of the model, we have re-run the model using historical value added of animal husbandry. For the two specific cases as mentioned in our previous manuscript, our modeled aggregate livestock loss for event (a) in 1996 in Yushu and Guoluo

prefectures was approximately 1.20 million heads/units (turned from 2.64 million sheep units modeled given the cattle-to-sheep ratio in Qinghai Province), and the historical record was 1.48 million[1] livestock recorded), and for event (b) in 1998 in Naqu Prefecture was 0.72 million head/unit (turned from 1.59 million sheep units), compared to 0.82 million livestock. Therefore, our model result did capture the loss of major events in specific regions, although it still suffered from uncertainty.

In our revision, we have added the temporal changes of livestock loss in section "*3.1.2 Model-derived annual snow disaster loss, 1980-2015*", in which model-derived historical losses were presented, and compared with recorded historical loss. In addition, model-derived loss of the two individual events were supplied to the second paragraph of section "4.1. Spatial patterns of livestock snow disaster risk in the QTP" (page 19 lines 21 – 28, clear version; or page 29 line 24 – page 30 line 5, track-change version)

Page 21 Line 8 "two critical indices": Is this presented in the Result section?
RE: The "two critical indices" were referring to disaster duration and growing season aggregate precipitation, two variables that critically determines livestock mortality rate in our vulnerability function.

In the revision, we have deleted this part per the comment of reveiwer#1 comment 12.

Page 22 Line 18: How the study can be applied for future? I consider that the method used here is not suitable when the climate is changing.
RE: Thank you for your question. In our modeling process, we have been trying to make our model framework capable of incorporating the changing climate and socioeconomic development. Following the existing work of climate change risk assessment (Carleton and Hsiang, 2016; Kinoshita et al., 2018; Tachiiri and Shinoda, 2012; Winsemius et al., 2016), our model consists of a set of response relationships: 1) a hazard module defines the relationship between daily weather condition and the occurrence (identification) of snow disasters; 2) a vulnerability function defines the relationship between even mortality rate and hazard intensity (duration, wind speed, growing season precipitation) and prevention capacity (as proxied by socioeconomic variable); 3) exposure in terms of herd size is used only in a multiplicative way to derive the final risk metrics in terms of sheep units.

In such a structure, climate condition and socioeconomic condition are merely inputs to our model, rather than a part of the model. Climate change will lead to changes in model input, and correspondingly model output. And that is why we want the model to have the capability to capture: in the short-term future, will the changing climate lead to more or less frequent snow disasters, with shorter or longer duration? Together with projected prevention capacity, will the mortality risk increase or decrease correspondingly? For the first question, our hazard module is capable of identifying/simulating snow disaster days based on climate inputs mimicking meteorological observers' decision using machine learning algorithm: given daily maximum, mean and minimum temperatures, precipitation, and maximum and mean wind speed, the module can exactly derive corresponding snow disaster event set. Applying it to future climate scenario can then generate future event set and investigate the change of disaster event
* * *
[1] The figure 1.08 million in the previous manuscript was from a literature, but unfortunately it is not consistent with those reported in the books. We have corrected it.

frequency and intensity (duration) in the future. For instance, in a warmer climate we may expect snow disaster events with less frequency and shorter duration.

In summary, climate change will not influence the model structure, but certainly it (model input) will change the model results. That is how the climate changing is taken into account in our model.

As reviewer #1 has raised similar comment (comment#3), we have refrained from claiming so strongly in the current manuscript but decided to conduct future risk assessment using the framework presented here in our next study.

[Technical corrections]

Page 1 Lines 20,22: 1/20a -> 20 years (also for all similar expressions).
RE: Thanks for pointing this out. We have corrected it throughout the manuscript and in the figures.

Pages 14-15 Fig 6: To be multi-colored like Fig 7 would be more reader-friendly.
RE: Thanks for the suggestion. We have updated the figures to use multi-color in the map.

Page 9 Line 7: "although unlikely" should be rephrased with better expression.
RE: Thanks for the suggestion. We have removed the phrase from the text.

Page 5 Lines 4-6: Hard to understand. Too many "and"s. "its needs" -> "it needs"? Delete one of the two "provide"s?
Page 10 Line 14: Fig. 3 -> Fig. 4?
Page 13 Line 2: topology -> topography?
Page 19 Line 22: There is no Table 2.
Page 19 Line 26 (also in Page 21 Line 28): higher -> longer.
Page 20 Line 8: Fig A2 -> A3?
RE: Above comments are related to typos. We have revised/corrected them as suggested.

[revised manuscript text omitted]
 identical dataset, we tried to include disaster prevention capacity as proxied by socioeconomic indicators into analysis, and followed Li et al. (2018)'s approach to derive the model with best predictive power. We tried different socioeconomic indicators including gross domestic production, value added of animal husbandry, fiscal revenue, fiscal expenditure, and gross domestic production per capita, following the suggestion from the literature (Wei et al., 2017) and one of the reviewer. We found the model using value added of animal husbandry yielded the best fitting result, having a deviance-based $R^2$ of 0.625 (More details of the model, including model fit statistics, response curves and model performance diagnostics, are provided in supplementary material S1). Therefore, following version of the generalized additive model was considered in further analysis,

$$\ln LR = s(Duration) + s(Wind) + s(P) + s(\sim\sim GDP\sim\sim Value\_Add) ,$$

(1)

where, livestock mortality rate induced by a snow disaster is determined by disaster duration (*Duration*), during disaster maximum daily mean wind speed (*Wind*), growing season (May-Sep) aggregate precipitation (*P*), and prevention capacity as measured by value added of animal husbandry (*Value_Add*) of the underlying county. *Duration* was used as the key indicator of hazard intensity. *Wind* and *P* were used to denote during disaster and pre-season environmental stressors, respectively (Li et al., 2018). *Value_Add* was used to indicate disaster prevention capability, which explicitly measures the size of animal husbandry, and implicitly represents prevention infrastructure and capability of risk management (Wei et al., 2017).

Given such a relationship, the vulnerability is a truly dose-response function between livestock mortality rate (mortality/herd size) and snow hazard intensity together with other environmental stressors and prevention capacity, as proposed by (Carleton and Hsiang, 2016). Different from simply defining vulnerability as the loss rate (Jongman et al., 2015; Kinoshita et al., 2018), the potential influence from socioeconomic development is embedded in the vulnerability function.

**2.2.3 Exposure**

Exposure measures the distribution of assets or population exposed to hazards (Kinoshita et al., 2018). In our framework, it should provide the spatial distribution of herd size exposed to snow disaster, and help turn the outputs from event loss modelling and livestock mortality rate (the response variable in the modelled vulnerability function) into mortality (death toll). According to its definition (Fernández-Giménez et al., 2012), livestock in nomadic grazing are prone to snow disaster the most as they obtain food mostly from grassland. Livestock raised in ranches or industrial livestock farms in agricultural regions, by contrast, are much less exposed as they have steady food supply from crop by-products and are well protected by infrastructure. Therefore, it is to estimate the number of livestock grazing on grassland to estimate livestock exposure to snow disaster.

A full gridded distribution map of herd size grazing on grassland in the QTP is not directly available, but it can be derived according to the rule-of-thumb for "forage-livestock balance". According to the *Forage-livestock Balance Management Approach* issued by the Ministry of Agriculture of China in 2006 to mitigate severe over-grazing in the pastoral areas of China (Shang et al., 2012), herd size grazing on grassland at the county level must be strictly controlled under carrying capacity. Therefore, a gridded carrying capacity map can be a good approximation of the actual herd size distribution exposed to snow disaster.

There are several factors deciding the carrying capacity of a given region, but the most important one is grassland type, according the *Ministry Standard of Calculation of Rangeland Carrying Capacity* issued by Ministry of Agriculture of China (NY/T 635-2015). In the standard, grassland type was used as the key identifier that essentially determines forage regrowth percentage, proper utilization rate of rangeland (of different grazing seasons), conversion coefficient of standard hay. Therefore, we tried to estimate the spatial distribution of exposure by turning grassland distribution data with the look-up

table for grassland-type to carrying-capacity relationship. For the look-up table, we adapted the plan of Xin et al. (2011) for Qinghai. For Tibet, we reviewed various criteria (Zhang et al., 2014) and  followed the official release of the Autonomous Region government  (Land Management Administration of Tibet Autonomous Region, 1994; Department of Agricultural and Pastoral of Tibet Autonomous Region, 2011)  The final look-up table was supplied in Appendix (Table A1). For grassland distribution, we used the Vegetation Map of the People's Republic of China (1:1 million) (Editorial Committee of Vegetation Map of China and Chinese Academy of Science, 2007), which offers detailed information about the spatial distribution of 11 vegetation type groups, 55 vegetation types, 960 plant formations, and more than 2000 dominant species in vector data. To match the look-up table and map information, we merged some vegetation types and used only the major grassland types (percentage area >0.5%) according to the survey from food and agriculture organization of the united nations  (FAO, 2005) (Fig. A2; Table A1). The estimated carrying capacity was aggregated to county-level and compared to the official release of Tibet Autonomous Region. The two datasets showed good agreement with a correlation coefficient of 0.769. Therefore, the estimated carrying capacity was used as exposure to turn mortality rate into mortaility.

**2.4 Loss modelling**

Snow disaster event losses measured with livestock mortality rate (death toll/ herd size) were modelled by taking requested inputs into the vulnerability function. *Duration* and *Wind* were outputs from the hazard module. Growing season aggregate precipitation *P* was computed from the climate forcing data.  Year-end county-level *Value_Add* were obtained from the statistical year books of Qinghai, Tibet, Sichuan, Gansu, and Xinjiang. County level *Value_Add* values were assigned to each grid within its boundary. When modelling losses, we considered two cases:

1) Loss based on historical prevention capacity: For modelling actual historical loss for model calibration and validation purpose, we used the actual county-level *Value_Add* of the study area during 1980-2015. As *Value_Add* increases along the time, it indicates the enhancing prevention capacity, and therefore reducing livestock mortality (rate) along the 
[revised manuscript text omitted]

---

## Author Response (AR2)

Dear Prof. Keiler

Thank you so much for your email. It is very glad to receive your minor revision request. We have read the reviewer's comments carefully, and responded piece by piece. Particularly, we want to ask for your suggestion about the last comment

"*it is better to present the historical change in observed loss too, rather than mentioning only statistical values*".

We would love to do so but the historical records are incomplete due to missing records in some years. We felt it might be misleading if we put them on the bar chart with continuous x-axis. However, if you do feel it is a must to put the historical records into Fig. 6, we will do so.

Again, we would thank you for our kind support in the submission and revision process. We benefited a lot from the discussion and did see substantial improvement of the quality of the article.

Warm regards,

Tao Ye

On behalf of the co-authors

**The manuscript has been revised carefully and I consider that it is almost ready for acceptance, except for some minor issues.**

RE: We would thank the reviewers for their comments that have helped us to substantially improve the paper. We have read the comments below carefully and responded point by point. Original comments were in bold.

**- I could not understand why "Using the annual loss approach cannot capture such changes" (last sentence of page 9 of nhess-2018-182-author_response-version3). My understanding is that we can assess the influence of increasing precipitation by annual analysis too.**

RE: In this sentence, "such changes" refers to the changes in frequency and intensity (duration) of snow disaster events. We intended to say that, if modeling loss based on annual-based analysis (as mentioned in Comment 1 of the reviewer on page 9, nhess-2018-182-author_response-version3), then we would be using variables on annual basis, i.e.. annual aggregate duration. Then frequency and intensity of snow disaster events will not be explicitly considered in the model.

**- For Fig R2, I agree that the estimated and the observed livestock number has good correlation, and this time I accept this approximation as a rough estimate. However, we should focus on the remaining residuals, particularly when data is above the y=x line (e.g., in case of +20 % residuals, that means overgrazing by 20%). I would like to emphasize that by taking this approximation we ignore one of the most significant issues in disaster risk management. I think if livestock is perfectly distributed following carrying capacity, large part of the problem in disaster risk management is solved.**

RE: Thank you very much for your comment. There are still uncertainties to use the carrying capacity to estimate actual herd size exposed. We totally agree with you that over-sized herds could largely increase the risk of mortality in snow disaster due to hay and fodder shortage. In response, we have added some discussion to address this issue in the revised manuscript. Please refer to lines 21-23, page 34: "*For the historical period, prior to the implementation of the forage-livestock balance policy, the actual herdsize exposed would be larger than carrying capacity due to over-grazing. In addition, having larger herdsize than the carrying capacity would exacerbate the pressure on grassland, lead to larger hay and fodder deficit in harsh winters, and increase herd vulnerability to snow disaster. Therefore, our model-derived historical loss was conservative.*"

**- "Worldwide, the QTP is a region that has most-suffered from livestock snow disasters..": Li et al (2018), saying "Climate, vegetation, and local nomadic pastoralism together make the QTP one of the most at-risk regions to suffer from livestock snow disaster." in Study Area part, may not be a suitable reference here. "most-suffered" seems to be their background, not their conclusion.**

RE: Thank you. We have changed to use (Shang, Gibb, & Long, 2012).

Shang, Z. H., Gibb, M. J., & Long, R. J. (2012). Effect of snow disasters on livestock farming in some rangeland regions of China and mitigation strategies - A review. *Rangeland Journal*, *34*(1), 89–101. https://doi.org/10.1071/RJ11052

**- There are two "3.1.2"s.**

RE: Thank you. This typo has been corrected.

**- In Fig 6, it is better to present the historical change in observed loss too, rather than mentioning only statistical values. In the orange case, using GDP at Purchasing Power Parity considered to be reasonable, do you use that?**

RE: Thank you very much for the comments.

1) We would love to present the observed loss in the Fig. 6. However, due to several years of missing records, presenting historical loss in a continuous bar chart could be misleading. I will try to discuss this issue with the editor to see the better solution.

2) GDP was replaced by the value added in animal husbandry after refitting the damage function (Eq. 1) according to the comment from Reviewer #1 in our last revision. We do not have the values in purchasing power parity, but we did turn the series into 2015 yuan.

[revised manuscript text omitted]

---

## Author Response (AR3)

Dear Prof. Margreth Keiler

Thank you so much for the message. It is a great news for us.

Per your request, we have included the historical loss records in Fig. 6 so a direct comparison is possible now. Missing data issue has been explained in the caption.

We have also double-checked the mathematical notation and English and house standards issues following the guidelines of NHESS.

Again, thank you for all your kind help during the process of revision the manuscript. We are ready for any further request from you and other editors to get the manuscript ready.

Cheers,

Tao

On behalf of the co-authors